# Prediction of plasma ctDNA fraction and prognostic implications of liquid biopsy in advanced prostate cancer

No consensus strategies exist for prognosticating metastatic castration-resistant prostate cancer (mCRPC). Circulating tumor DNA fraction (ctDNA%) is increasingly reported by commercial and laboratory tests but its utility for risk stratification is unclear. Here, we intersect ctDNA%, treatment outcomes, and clinical characteristics across 738 plasma samples from 491 male mCRPC patients from two randomized multicentre phase II trials and a prospective province-wide blood biobanking program. ctDNA% correlates with serum and radiographic metrics of disease burden and is highest in patients with liver metastases. ctDNA% strongly predicts overall survival, progression-free survival, and treatment response independent of therapeutic context and outperformed established prognostic clinical factors. Recognizing that ctDNA-based biomarker genotyping is limited by low ctDNA% in some patients, we leverage the relationship between clinical prognostic factors and ctDNA% to develop a clinically-interpretable machine-learning tool that predicts whether a patient has sufficient ctDNA% for informative ctDNA genotyping (available online: https://www.ctDNA.org). Our results affirm ctDNA% as an actionable tool for patient risk stratification and provide a practical framework for optimized biomarker testing.

Metastatic castration-resistant prostate cancer (mCRPC) is lethal and globally results in over 375,000 deaths annually[1]. Although new treatments have extended survival, clinical outcomes remain heterogeneous, ranging from relative indolence to upfront therapy resistance and rapid death[2]. Prognostic classification schemes and nomograms leveraging clinical and routine laboratory prognostic factors are used to estimate disease aggression and influence patient management[3-8]—including treatment selection and intensification, choice of early versus late chemotherapy[9], as well as imaging frequency and clinical trial prioritization[10]. However, existing prognostication strategies provide modest stratification value and were developed using historical trial datasets that do not represent the contemporary mCRPC population that receives several lines of systemic therapy and has a median survival of more than 2–3 years.

In patients with cancer, tumor DNA is shed into the blood and mixes with normal cell-free DNA (cfDNA) from apoptosed leukocytes[11]. ctDNA fraction (ctDNA%) is the proportion of tumor-derived cfDNA[12] and is an emerging prognostic factor across cancers and clinical scenarios[13,14]. Early studies suggested that high ctDNA% is associated with poor prognosis in mCRPC and may be more accurate than existing clinical prognostic factors[14-22]. However, the precise relationship between ctDNA%, established clinical prognostic factors (including serum markers and radiographic features), subsequent therapy response, and overall life expectancy is unknown. Evaluating these relationships requires large standardized cohorts to determine whether ctDNA% testing can outperform existing prognostication strategies and warrants incorporating into clinical practice. Excitingly, ctDNA% is increasingly reported on commercially-available tests that genotype ctDNA to determine treatment-predictive biomarker

✉e-mail: matti.annala@tuni.fi; kchi@bccancer.bc.ca; awwyatt@mail.ubc.ca

status[23], meaning that ctDNA%-prognostication is poised to rapidly influence patient management pending its clinical validation.

In current clinical practice, plasma ctDNA testing is typically used to identify actionable somatic alterations (i.e., cancer genotyping), and tests can be ordered multiple times across sequential disease progression on different systemic therapies. In particular, ctDNA genotyping is used in patients with mCRPC to evaluate eligibility for poly (ADP-ribose) polymerase [PARP] inhibitor treatment[24,25]. However, ctDNA% is critical for interpreting genotyping results since low ctDNA% is common and precludes detection of clinically-relevant alterations such as somatic *BRCA2* truncating mutations and homozygous deletions[20,26]. Healthcare providers are increasingly aware of this limitation of ctDNA testing and there is considerable debate about the reliability of results and when it is necessary to rely instead upon tumor tissue testing for cancer genotyping[27,28]. Unlike ctDNA%, which can be measured with inexpensive assays such as low-pass whole-genome sequencing[29], comprehensive ctDNA genotyping requires relatively expensive deep sequencing. Ideally, tests for ctDNA genotyping should be directed towards patients with a high probability of abundant ctDNA, while patients likely to have lower ctDNA% could be prioritized for tissue-based or germline-only biomarker testing. There are currently no practical tools to estimate ctDNA% sufficiency for genotyping prior to blood draw, resulting in treatment delays as patients await potentially uninformative ctDNA genotyping results, and wasted resources in profiling samples with insufficient tumor material.

In this context, we have assembled a large standardized metacohort of mCRPC patients with serial cfDNA linked to comprehensive time-matched clinical annotation. We dissect the complex associations between ctDNA%, clinical prognostic markers, and treatment outcomes, and build a public user-friendly tool (http://ctDNA.org) to predict ctDNA% from routine clinical markers to help clinicians prioritize patients for ctDNA genotyping.

## Results

### Metacohort clinical characteristics and ctDNA fraction distribution

We profiled 738 plasma cfDNA samples from 491 clinically-progressing mCRPC patients with comprehensive clinical annotation matched to times of blood collection (Fig. 1A, B; Supplementary Figs. 1, 2; Supplementary Data 1). This metacohort is comprised of previously published samples and data from 292 patients enrolled in two completed randomized multicentre phase II trials addressing treatment involving standard-of-care drugs for first- and second-line mCRPC (NCT02125357 and NCT02254785)[16,20,30] and new samples and data from 199 previously-unpublished patients from a prospective province-wide plasma cfDNA biobanking program. All clinical annotation and endpoints have been standardized (Methods) and we have obtained updated outcomes for consenting trial patients. Overall, 463 (94%), 213 (43%), and 62 (13%) patients provided cfDNA samples within 1 month prior to first (i.e., baseline), second, or third-line mCRPC therapy, respectively, and 206 (42%) provided ≥2 timepoints. Time-matched clinical characteristics (including Eastern Cooperative Oncology Group performance status and serum laboratory measurements of disease burden) were broadly consistent across successive lines of treatment and mirrored contemporary real-world mCRPC populations (Fig. 1C–H; Table 1). Line-specific treatment patterns also reflected standard clinical practice, with most patients receiving AR-axis inhibitors abiraterone or enzalutamide for first-line mCRPC[31,32]. Notably, 17% (*n* = 81) patients initiating first-line mCRPC therapy received prior systemic treatment intensification for castration-sensitive disease—typically via addition of docetaxel chemotherapy (74/81; 91%) rather than AR pathway inhibitors (7/81; 9%) to continuous androgen deprivation therapy (ADT)—reflecting standard practice at time of enrollment

(Methods). Consequently, most patients in our metacohort were naive to standard first-line therapies for mCRPC[30]. Median follow-up for overall survival (OS) measured from initiation of first, second, and third-line mCRPC therapy was 20.3 (range: 0.4–81.6), 14 (0.5–63.5), and 10.6 (2.2–32.8) months, respectively. Median OS measured from first-line mCRPC treatment initiation was 25 months (95% CI: 22.4–28.2) (Fig. 1B; Table 1).

For all new plasma samples, we performed deep targeted sequencing of plasma cfDNA and leveraged multiple lines of evidence to estimate ctDNA%, including copy number alterations and somatic mutation allele frequencies corrected for outliers and concomitant loss-of-heterozygosity (Methods; Supplementary Fig. 3). For previously published samples, ctDNA% was re-estimated from existing sequencing data using the same approach. Concurrent deep sequencing of patient-matched white blood cells (performed for all patients) minimized the likelihood of false-positive ctDNA% estimates due to germline or clonal-hematopoiesis variants[26,33]. Across all samples, median ctDNA% was 5.0% (range: 0–89.2%; IQR: 0–25.2%) and 63.8% had evidence for ctDNA with our relatively conservative detection thresholds. We partitioned ctDNA% into categories of high (30–100%), low (2–30%), and undetected ctDNA (<2%)—the proportion of patients with high ctDNA% was similar across all lines of therapy (Table 1; Fig. 1E, F).

To characterize ctDNA% temporal dynamics across serial progression events, we analyzed the 227 consecutive same-patient sample pairs in our metacohort (median collection interval: 6.5 months; range: 0.9–32.5; IQR: 3.7–11.2). Serial ctDNA% measurements were correlated (Pearson *R* = 0.70, *p* < 0.0001), especially for progression samples collected <8 weeks apart (*R* = 0.97, *p* < 0.0001) consistent with primary therapy resistance (Fig. 1I, J). Low ctDNA% prohibits comprehensive assessment of all classes of ctDNA biomarkers (especially copy number alterations)[26]. In the event of a poorly informative first collection (i.e., ctDNA < 2%), we investigated whether later re-testing can overcome an initially low ctDNA%, considering conventional limits of detection for mutations (1–2% ctDNA) and copy number deletions (≥30%) used by commercial tests that genotype ctDNA[23,34]. ctDNA ≥ 2% at a later progression timepoint was significantly less likely if the earlier sample was ctDNA < 2% (30.6% versus 81.7% if initially positive, odds-ratio = 0.09; Fisher's Exact Test *p* < 0.001) (Fig. 1J). Strikingly, only 8.7% of sample pairs with initial ctDNA < 30% increased to ≥30% in a subsequent collection, and only two patients (2.2%) converted from <2% to ≥30%. Probability of ctDNA% conversion between baseline and second-line was related to depth of PSA response on first-line therapy, with consecutively increasing ctDNA% occurring more frequently in patients who did not achieve a PSA response >50% (*p* < 0.01) (Fig. 1K).

### ctDNA fraction correlates with clinical metrics of tumor aggression

Previous studies in multiple cancer types have demonstrated that plasma ctDNA abundance is positively correlated with metastatic volume[35–41], and that distinct patterns of metastasis may influence patient prognosis[42–45]. We correlated baseline ctDNA% with 15 disease associated features including diagnostic and time-matched serum and radiographic variables with prognostic relevance in mCRPC[7]. ctDNA% was significantly elevated in patients with liver metastases on conventional imaging (median ctDNA 42% versus 4.9% in patients with lymph node-only disease, Mann-Whitney U (MWU) *p* < 0.001), detected in 90% of patients compared to only 59% of patients with bone metastases (without visceral involvement) and 57% with lymph node-only disease (Fig. 2A). Consistent with this observation, serum lactate dehydrogenase (LDH)—a reliably demonstrated negative prognostic factor for mCRPC that is classically associated with liver metastases[46]—was also strongly correlated with ctDNA% (Spearman *r* = 0.41, *p* < 0.0001) (Fig. 2B)[7,18,20]. In patients with bone metastases,

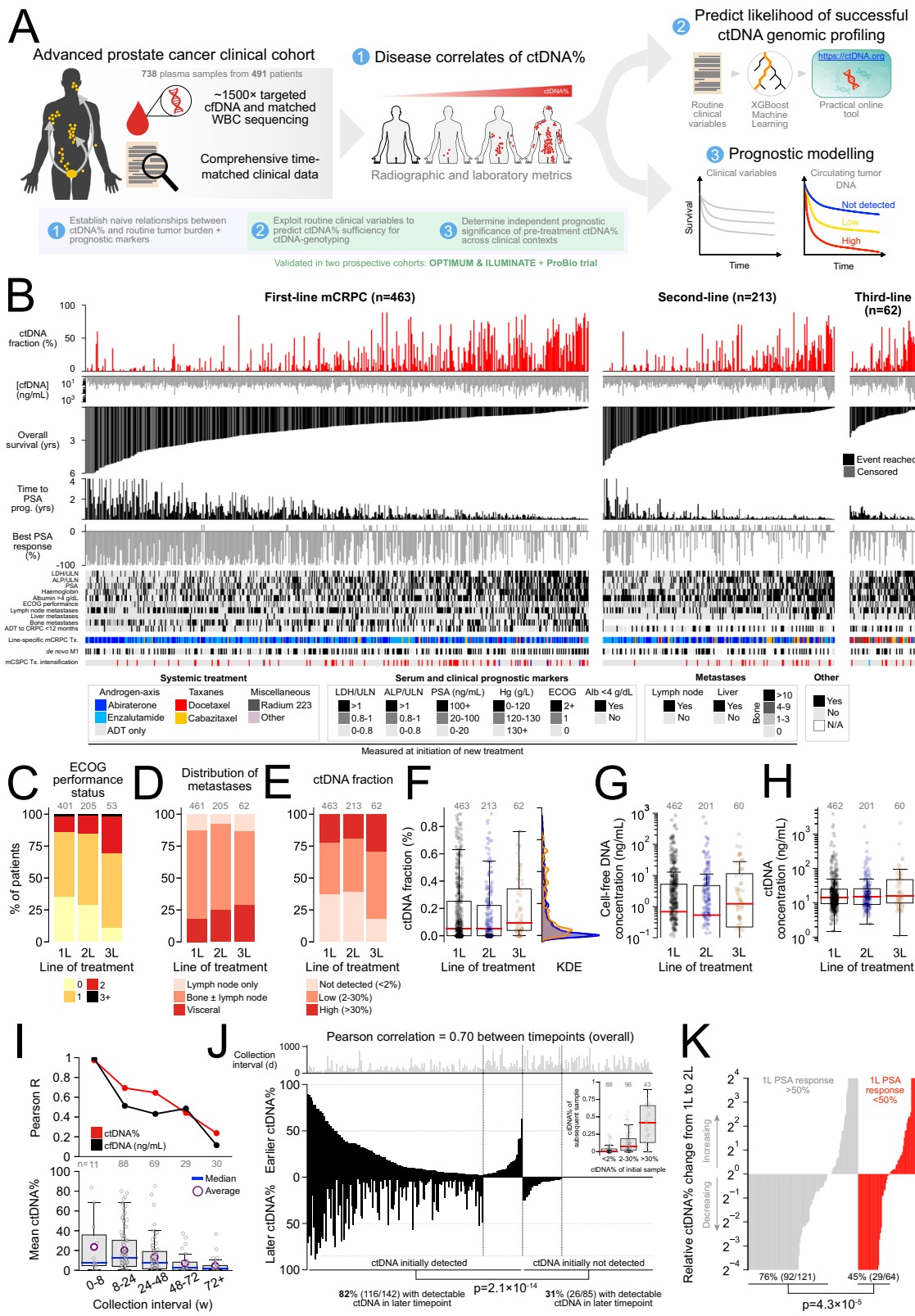

≥10 lesions on bone scan was associated with 2.7-fold higher ctDNA% than <10 bone lesions (median ctDNA 10.4% versus 3.9%, MWU $p < 0.001$), corroborating prior data showing elevated ctDNA% in patients with higher disease burden[36,37,40,41]. Among plasma markers, ctDNA% was most strongly correlated to cfDNA concentration ($r = 0.55$), although most blood analytes (including cfDNA concentration and LDH) were also inter-correlated (Fig. 2B, C). cfDNA

concentration was similarly correlated with most aforementioned clinical factors, although the effect size was weaker relative to ctDNA % (Supplementary Fig. 4; Fig. 2C).

Several clinical hallmarks of aggressive disease in the castrate-sensitive setting—including diagnosis of de novo metastatic cancer and short time to mCRPC progression from ADT initiation (<12 months)—were also linked to higher baseline mCRPC ctDNA%, reflecting a

**Fig. 1 | Clinical mCRPC cohort with comprehensive clinical annotation and ctDNA-fraction estimates. A** Study overview. **B** Per-patient summary of clinical prognostic metrics and treatment outcomes stratified by line of mCRPC therapy, illustrating approximate relationships between high ctDNA-fraction (ctDNA%) (see Supplementary Data 2) and poor prognosis (see Supplementary Data 1 for complete list of clinical variables). All variables (including ctDNA%) are measured at time of line-specific mCRPC treatment initiation except for pre-mCRPC clinical history and diagnostic metrics (i.e., time from androgen deprivation therapy (ADT) initiation to CRPC diagnosis, de novo metastatic diagnosis, and treatment intensification for metastatic castration-sensitive prostate cancer (mCSPC)). Bars representing right-censored time-to-event clinical endpoints are colored gray (events reached are black). Patients whose best PSA response was rising PSA (i.e., nadir at baseline) have been truncated at a fixed positive value. Note that bone metastases were only enumerated in the first-line context, although all patients (independent of treatment line) were evaluated for bone lesion presence/absence. Temporal consistency

of key patient clinical characteristics (**C**, **D**) and plasma cfDNA and ctDNA measurements (**E**–**H**) per initiating line of treatment. Number of patients with evaluable data matched to line of treatment annotated above (see Supplementary Data 3). **I** Correlation between consecutive same-patient ctDNA% and cfDNA concentration measurements taken at sequential clinical progressions as a function of collection interval. **J** Mirrored barplot showing same-patient ctDNA% across 227 consecutively collected cfDNA sample pairs (*p*-value reflects Fisher's Exact Test). In-set boxplot is centered at median and displays interquartile ranges (IQR) and minima and maxima extending to 1.5× IQR. **K** Serial ctDNA% dynamics are associated with PSA response on intervening treatment. Relative ctDNA% change from initiation of first-line mCRPC therapy to initiation of second-line therapy in patients who did or did not achieve a PSA response ≥50% to first-line treatment. Fisher's Exact Test compares proportion of patients in each category with serially decreasing ctDNA%. All *p*-values are two-sided. Yrs years, Tx treatment, PCa prostate cancer, prog. progression, w weeks, KDE kernel density estimation, ULN upper-limit of normal.

**Table 1 | Cohort clinical characteristics measured per line of mCRPC therapy[a]**

| Patient characteristic | First line (n = 463) | Second line (n = 213) | Third line (n = 62) |
|---|---|---|---|
| Age at initial prostate cancer diagnosis (years) | 68 (41–96) | 67 (41–86) | 63 (41–75) |
| Gleason Grade Group 4–5 | 71% (245) | 71% (96) | 71% (32) |
| De novo metastatic diagnosis | 46% (171) | 49% (69) | 42% (19) |
| Treatment intensification for mCSPC | 17% (81) | 11% (25) | 13% (8) |
| Taxane chemotherapy (docetaxel) | 16% (74) | 11% (25) | 12% (7) |
| Abiraterone, enzalutamide | 2% (6) | 0% (0) | 0% (0) |
| Other | <1% (1) | 0% (0) | 0% (0) |
| CRPC within 12 months of ADT initiation | 40% (174) | 53% (104) | 57% (32) |
| Age at systemic mCRPC treatment initiation (years) | 73 (45–98) | 73 (45–93) | 69 (45–88) |
| Systemic treatment for mCRPC | 100% (463) | 100% (213) | 100% (62) |
| Taxane chemotherapy (docetaxel or cabazitaxel) | 8% (39) | 17% (37) | 49% (30) |
| Abiraterone, enzalutamide | 91% (420) | 78% (166) | 11% (7) |
| Other | 1% (4) | 5% (10) | 40% (25) |
| ECOG performance status 0–1 | 86% (345) | 70% (143) | 70% (37) |
| Alkaline phosphatase > ULN | 35% (155) | 34% (73) | 45% (27) |
| Lactate dehydrogenase > ULN | 23% (93) | 22% (45) | 36% (19) |
| Hemoglobin (g/L) | 130 (79–174) | 128 (79–157) | 127 (97–156) |
| PSA (ng/mL, plasma) | 26 (0–5800) | 19 (0.2–1604) | 53 (3.8–812) |
| Visceral metastases | 18% (83) | 24% (51) | 29% (18) |
| Lymph-node only metastases | 12% (55) | 7% (15) | 10% (8) |
| Cell-free DNA concentration (ng/ml) | 14 (1.5–3870) | 15 (2.4–1650) | 16 (1.1–2140) |
| ctDNA fraction | 5% (0–89) | 5% (0–89) | 10% (0–77) |
| ctDNA concentration (ng/mL) | 0.7 (0–3146) | 0.5 (0–771) | 1.14 (0–1286) |
| ctDNA not detected (i.e., <2%) | 38% (174) | 39% (83) | 18% (11) |
| ctDNA fraction 2–30% | 40% (187) | 42% (90) | 53% (33) |
| ctDNA fraction >30% | 22% (102) | 19% (40) | 29% (18) |
| Follow-up for OS (months) | 20 (0.36–81.6) | 14 (0.52–63.5) | 11 (2.16–32.8) |
| Median OS (months)[b] | 25 (22.4–28.2) | 15.7 (13.6–17.8) | 11.1 (8.7–14.4) |

[a]Data are median (range), or % (n); percentages reflect proportion of patients with complete data for the given variable.
[b]95% confidence interval.
*ADT* androgen deprivation therapy, *mCSPC* metastatic castrate-sensitive prostate cancer, *ECOG* Eastern Cooperative Oncology Group, *GGG* Gleason Grade Group, *mCRPC* castration-resistant prostate cancer, *OS* overall survival, *PSA* prostate-specific antigen, *ULN* upper limit of normal.

continuity in disease aggression over the clinical spectrum of advanced prostate cancer. By contrast, PSA concentration at diagnosis and Gleason Grade Group were poorly or not correlated with baseline mCRPC ctDNA% (Fig. 2A, B) regardless of diagnosis of de novo or metachronous metastatic disease ($p = 0.71$; two-way ANOVA test for interaction). Patients who received treatment intensification for mCSPC (versus ADT monotherapy) had a 1.9 × higher baseline mCRPC

ctDNA%, likely reflecting patient selection for higher-volume and/or poorer prognosis disease[47].

## Machine-learning model to predict utility of ctDNA somatic genotyping

No practical tools exist to predict ctDNA% and determine at or before blood draw whether ctDNA genotyping will be informative, or whether

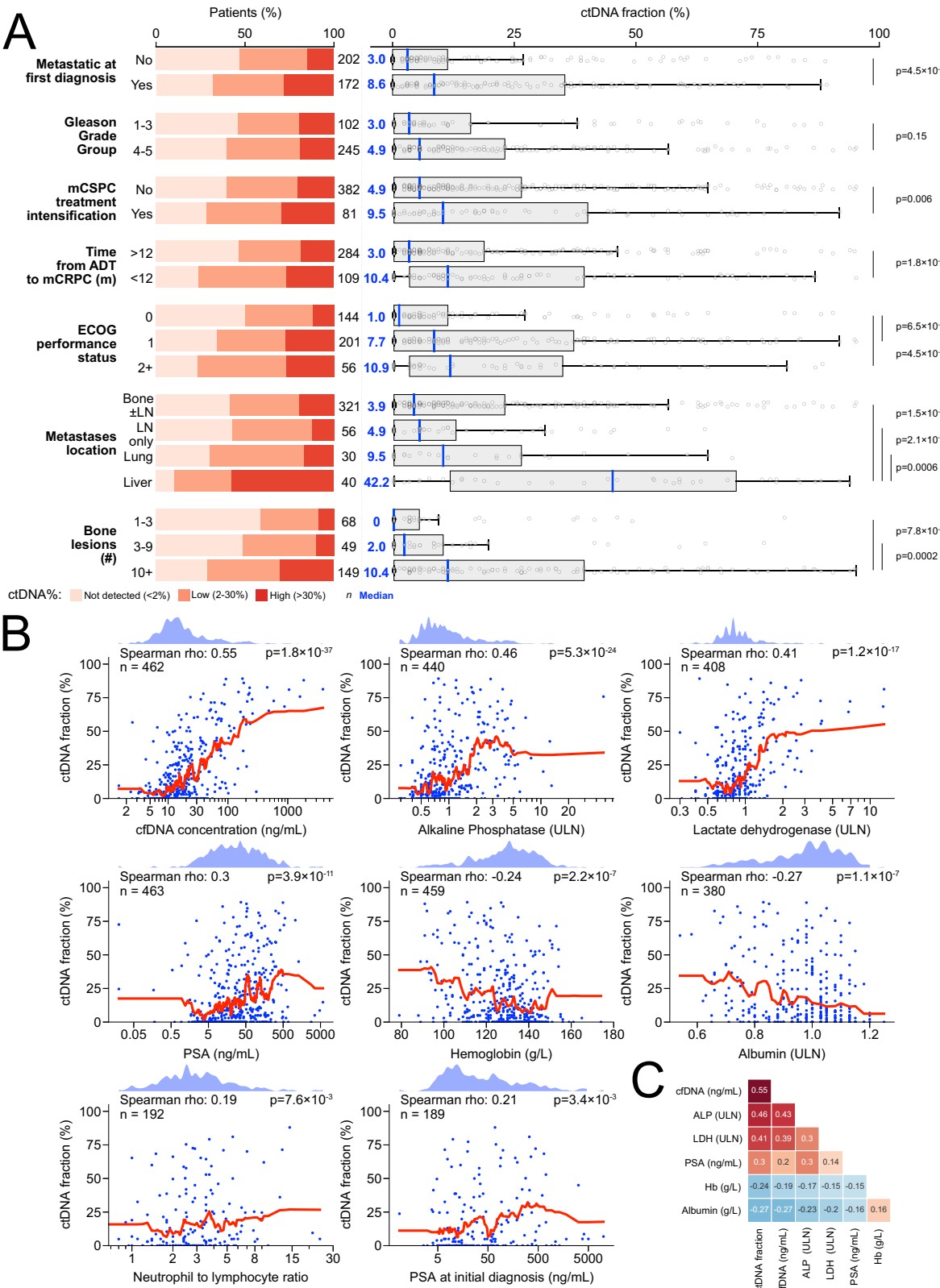

tissue-based genotyping should be a preferred testing strategy[38,48]. We reasoned that the modest correlations between ctDNA% and clinical factors elucidated above (Fig. 2) could be integrated into a single more powerful predictive tool, helping providers to decide which patients to prioritize for ctDNA- versus tissue-genotyping (based on anticipated likelihood for sensitive characterization of somatic alterations). Therefore, we developed a clinically-interpretable machine-learning

model to predict ctDNA% from 17 laboratory and radiographic clinical features as well as diagnostic characteristics. We trained a gradient-boosted tree (XGBoost) model using all 463 baseline cfDNA samples, with hyperparameter tuning via 5-fold cross-validation. Model accuracy was estimated via 20-fold cross-validation (Methods). We optimized the XGBoost model for binary classification of samples as either above or below 2% ctDNA: the approximate lower limit of detection for

**Fig. 2 | Serum and radiographic prognostic clinical features correlate with baseline ctDNA fraction. A** Fraction of patients with ctDNA>30%, ctDNA 2-30%, and ctDNA<2% (left) and ctDNA% as a continuous variable (right) across various categorical clinical subgroups. Note that the "bone ± lymph node" category excludes patients with visceral metastases, and the "lung" category excludes patients with liver metastases; the "liver" category does not exclude any other metastatic subgroup. *P*-values reflect Mann-Whitney U tests and are two-sided; boxplots are centered at the median and display interquartile ranges (IQR) and minima and maxima extending to 1.5× IQR. **B** Correlation between ctDNA% and eight continuous prognostic serum markers. K-nearest neighbor regression (neighbors = 20 with uniform weights; red line) is used to nonparametrically visualize each bivariate relationship (i.e., avoids making assumptions about how ctDNA% is linked to each clinical factor). Kernel density estimates shown above. Spearman *p*-values are two-sided. **C** Correlation matrix showing that most serum prognostic markers are co-correlated. Spearman's rho is annotated. See Supplementary Data 2 for per-patient ctDNA% values. mCSPC metastatic castration sensitive prostate cancer, ADT androgen deprivation therapy, m months, LN lymph node, Hb hemoglobin, PSA prostate-specific antigen, ULN upper limit of normal.

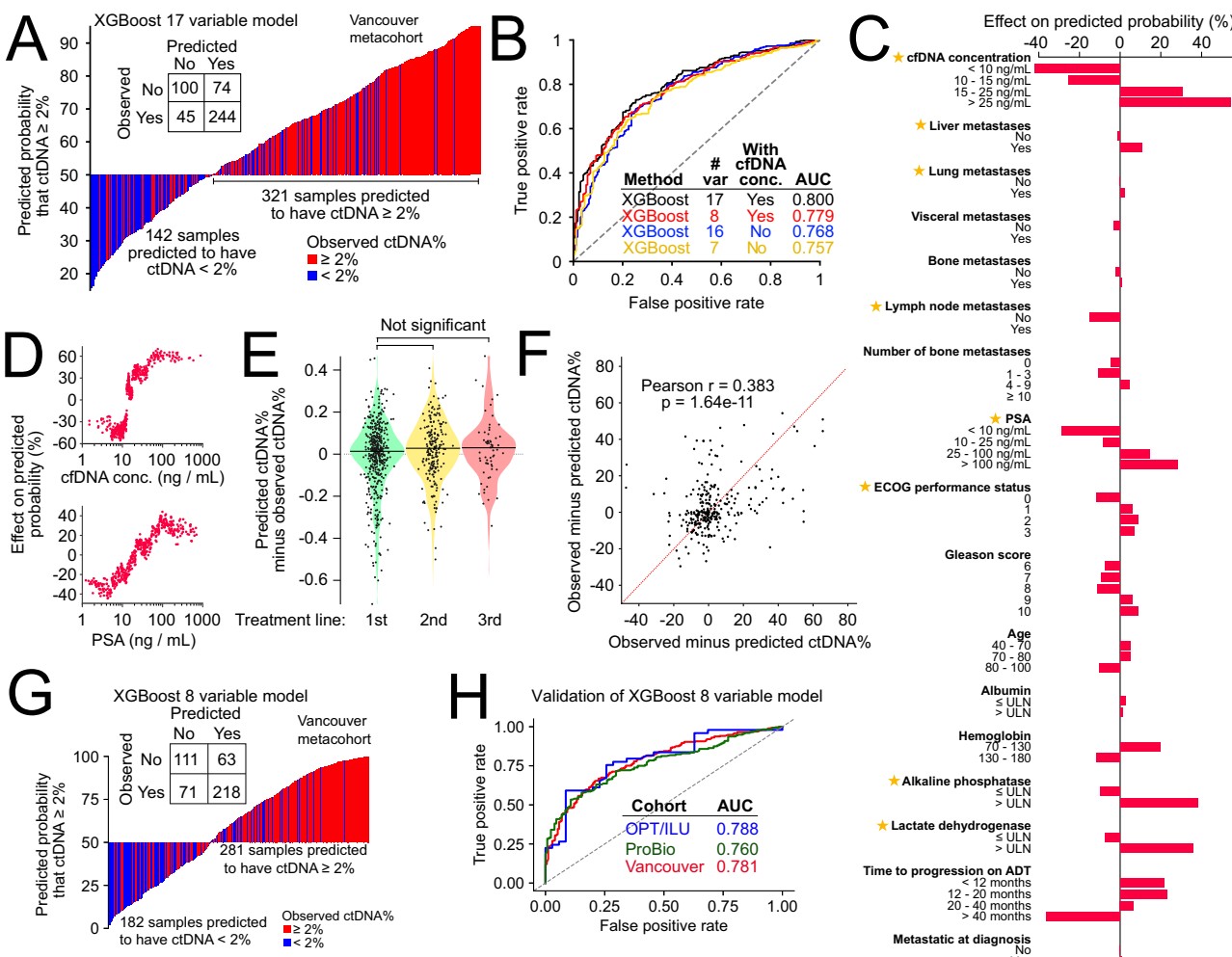

**Fig. 3 | ctDNA fraction prediction based on routine clinical variables.**
**A** Predicted probability of ctDNA≥2% based on our 17-feature XGBoost model applied to 463 first-line mCRPC samples (see Supplementary Data 6 for complete list of clinical variables used for model training plus model performance metrics). True observed ctDNA ≥ 2% status is indicated with color. In-set confusion matrix for classification of ctDNA ≥ 2%. **B** Receiver operating characteristic curves for four separately trained and optimized XGBoost models evaluating different sets of clinical input features. **C** Average contribution of individual clinical input features to model predictions, quantified using Shapley (SHAP) values (evaluated on the 17-feature XGBoost model). Stars indicate clinical features selected for the parsimonious 8-variable ctDNA% prediction model. **D** SHAP scores for cfDNA concentration and PSA as continuous variables (evaluated on the 17-feature XGBoost model). **E** Uniform model prediction error across sequential lines of mCRPC treatment (pairwise comparisons use the Mann-Whitney U test and are not corrected for multiple hypothesis testing). **F** Scatterplot showing observed minus predicted ctDNA% in the earlier (x-axis) versus later (y-axis) timepoint for 288 same-patient sample pairs across different lines of treatment (*p*-value is two-sided). Positive correlation between axes suggests the existence of patient and/or tumor-specific multipliers on ctDNA% (i.e., clinical or biological variables not accounted for in our model). **G** Predicted probability of ctDNA ≥ 2% based on our 8-feature XGBoost model applied to 463 first-line mCRPC samples. **H** Validation of our 8-feature XGBoost model in two external clinical trial cohorts. var variables, conc. concentration.

point mutations and indels in contemporary genotyping assays using large targeted panels[23,34]. Our proof-of-principle model achieved relatively high accuracy, as measured by area under the curve of receiver operating characteristics (AUC = 0.80) (Fig. 3A, B). Performance was orthogonally validated using a dimensionally-weighted K-nearest neighbor classification model trained on the identical 17-feature set (AUC = 0.74, Fig. 3B; Supplementary Fig. 5A).

Consistent with prior bivariate rank correlations (Fig. 2), cfDNA concentration had the highest relevance for predicting ctDNA ≥ 2%, whereas features associated with initial prostate cancer diagnosis (i.e.,

Gleason grade and stage at diagnosis) were less informative (Fig. 3C, D). Importantly, we observed equal accuracy of our model when applying it to 275 second- and third-line samples it had not encountered during training, suggesting generalizability across mCRPC clinical contexts (Fig. 3E). We next utilized our XGBoost model to understand whether there exist any unaccounted factors causing a consistent bias in observed ctDNA% relative to a patient's model-predicted ctDNA%. After analyzing 200 patients with multiple cfDNA samples, we found that patients with greater-than-predicted ctDNA% in one cfDNA sample also showed a greater-than-predicted ctDNA% in their other samples (Pearson $r = 0.38$, $p < 0.0001$), suggesting additional patient- or tumor-specific determinants of ctDNA% (Fig. 3F).

Recognizing that comprehensive and standardized clinical annotation is not always available in real-world settings, we also developed a more parsimonious XGBoost classification model focused on seven routinely collected clinical features (plus cfDNA concentration) that had high importance for ctDNA% classification and reflected time-matched information. Specifically, we excluded diagnostic features whose predictive relevance is expected to gradually fade in later disease stages (e.g., Gleason grade) or outcomes features that are confounded by rapidly shifting standard of care (e.g., time from start of ADT to mCRPC)[49]. Since this secondary model achieved similar performance (including cfDNA concentration as input: AUC = 0.78, PPV = 0.75, NPV = 0.63; excluding cfDNA concentration: AUC = 0.76, PPV = 0.78, NPV = 0.55), we implemented it as a practical point-of-care online tool (https://www.ctDNA.org) (Supplementary Data 6). Importantly, our tool is trained to handle every combination of missing input variables (i.e., $n = 255$ combinations) and produces a best-effort numerical prediction of per sample ctDNA%, empowering users to initiate ctDNA biomarker testing conditional on custom objectives (e.g., priority of mutation versus copy number information; tolerance for likelihood of false negatives). We validated the performance of our parsimonious 8-feature model in two external prospective mCRPC datasets collectively including 391 patients with first-line mCRPC, achieving similar AUCs for predicting ctDNA ≥ 2% of 0.76–0.78 (Methods; Fig. 3G, h, Supplementary Fig. 6, Supplementary Data 6). Patient clinical characteristics for one of the two validation cohorts ($n = 81$ patients) has been published previously[50].

### ctDNA fraction strongly predicts clinical and biochemical outcomes

In patients where ctDNA genotyping has been performed, prior studies indicate that measured ctDNA% alone (i.e., independent of any identified genomic driver alterations) is prognostic for outcomes. We sought to validate and expand upon prognostic trends identified in smaller studies with limited follow-up for outcomes (including our own previously published work on two prospective trial cohorts included in our metacohort[16,20]). We correlated ctDNA% to OS, PSA-PFS, and PSA response rates in the context of first and second-line mCRPC, incorporating treatment context and known clinical prognostic factors in secondary multivariable analyses (Fig. 4A–G; Supplementary Fig. 7). Notably, OS and PSA-PFS event rates in our metacohort were 70% and 75% for first-line, and 84% and 83% for second-line, reflecting the long median follow-up of our metacohort (Table 1; Supplementary Data 3).

Patients with high baseline ctDNA% (>30%) had a 5 times greater risk of PSA progression on first-line treatment (95% CI 3.7–7.0, p < 0.001) (Fig. 4B, E) and 5.6 times greater risk of death (95% CI 4.1–7.6, $p < 0.001$) (Fig. 4A, D) versus patients with ctDNA < 2%. Consistent with these observations, increasing baseline ctDNA% was linked to incrementally lower first-line PSA response rates (Fig. 4G). When collected prior to initiation of second-line mCRPC therapy, ctDNA% was similarly associated with second-line PSA response rates, time to PSA progression, and OS (measured from second-line treatment initiation) (Supplementary Fig. 7). In multivariable analyses adjusting for 8 established clinical prognostic markers (each significant in univariable testing), high ctDNA% remained independently associated with PSA-PFS and OS in both a first- and second-line treatment context, consistently outperforming other clinical covariates (Fig. 4C, F; Supplementary Data 5). ctDNA% was also prognostic for OS and PSA-PFS as a continuous variable (HR = 1.03 [95% CI: 1.02-1.03], $p < 0.001$; HR = 1.02 [95% CI: 1.02-1.03], $p < 0.001$; hazard ratios reflect 1% unit increase in ctDNA%). These data demonstrate that ctDNA% is strongly and independently prognostic across clinical scenarios.

We additionally tested whether ctDNA concentration (i.e., nanograms of ctDNA per mL plasma, the product of total cfDNA concentration and ctDNA%) enabled more precise prognostication than ctDNA% or cfDNA concentration alone. When dichotomized by median, baseline ctDNA% and ctDNA concentration were associated with comparable univariable hazard ratios for OS (HR = 3.18 [95% CI: 2.53–3.99], $p < 0.001$; HR = 3.28 [95% CI: 2.61–4.12], $p < 0.001$) and both enabled superior patient stratification relative to cfDNA concentration (HR = 2.05 [95% CI: 1.64-2.56], $p < 0.01$) (Fig. 4A; Supplementary Fig. 8)[51]. ctDNA% was more strongly prognostic than cfDNA concentration (both variables dichotomized at median) independent of treatment line and endpoint (Fig. 4A, B; Supplementary Fig. 8; Supplementary Data 4). Finally, we observed that incremental increases in baseline ctDNA% were associated with greater relative increase in risk of death when ctDNA% was low, implying that the relationship between ctDNA% and risk is nonlinear (Fig. 4H).

## Discussion

We systematically dissect the relationship between ctDNA%, synchronous laboratory and radiographic prognostic indices, and clinical outcomes in the largest such standardized metacohort of mCRPC patients to date. We demonstrate that ctDNA% is powerfully prognostic for multiple validated clinical endpoints, independent of treatment context and recognized clinical prognostic covariates in mCRPC. Leveraging these correlative insights, we build a practical point-of-care machine-learning framework to predict the likelihood of informative ctDNA genotyping prior to blood collection. Our work nominates a hypothetical biomarker testing framework enabling clinicians to decide between ctDNA testing and alternative testing modalities (e.g., archival tissue, fresh tissue biopsy, or germline-only) on the basis of anticipated ctDNA% (Fig. 5).

Our data, together with prior smaller studies, authenticate ctDNA% as a comprehensive prognostic tool across the clinical spectrum of mCRPC[14–18,20–22]. ctDNA% was linked to multiple clinical metrics of tumor burden and disease aggression (Fig. 1B; Fig. 2), although the effect size between ctDNA% and any individual marker (including PSA) was moderate at best ($R < 0.55$), illustrating that ctDNA% is not merely a surrogate for existing prognostic indices. This is reinforced by two additional observations: (1) ctDNA% remaining highly prognostic for time-to-event outcomes after adjustment for known prognostic features in multivariable models (Fig. 4A–F), and (2) the imperfect performance (AUC: 0.77–0.80) of our machine-learning models for predicting ctDNA ≥ 2% from clinical characteristics (Fig. 3). Collectively, these data demonstrate that ctDNA% captures unique biology and offers an additional dimension of prognostic information. Future clinical trial designs for advanced disease should consider incorporating ctDNA% as a stratification factor for randomization, and/or evaluating ctDNA% imbalances between arms to facilitate *post hoc* interpretation. New studies investigating additional determinants of ctDNA% should utilize next-generation targeted imaging (e.g., [68Ga]PSMA-PET/CT in prostate cancer) for more precise quantification of disease burden and location[40]—as well as investigate the potential relevance of tumor cell proliferation indicators (e.g., Ki-67-positive tumor nuclei or total lesion glycolysis) and microenvironmental factors (e.g., tumor vascularization, macrophage infiltration) on ctDNA%.

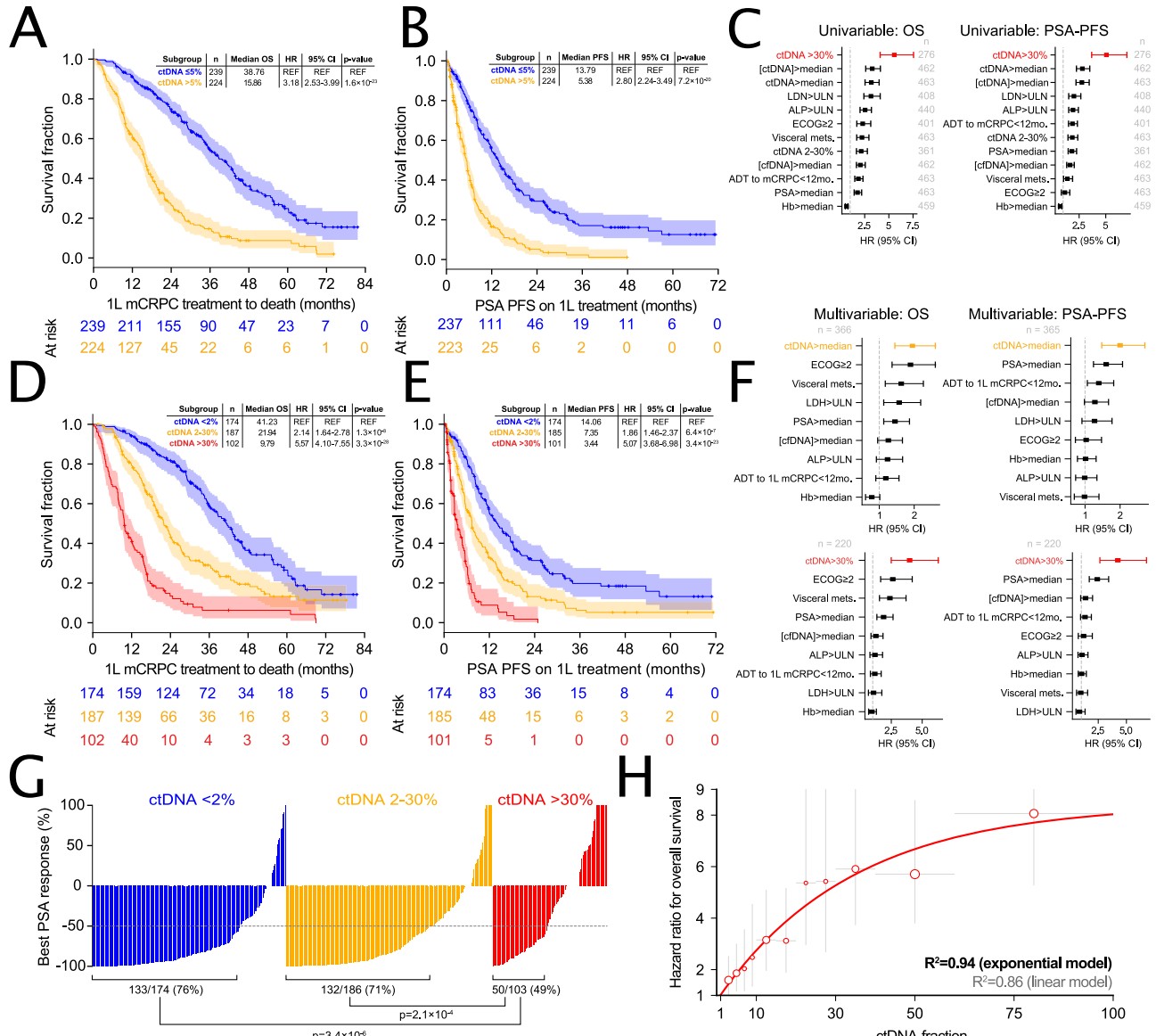

**Fig. 4 | ctDNA fraction is independently associated with mCRPC treatment outcomes.** Kaplan-Meier estimates of time from initiation of first-line systemic therapy for mCRPC to death or last follow-up (**A** + **D**) and PSA progression-free survival on first-line therapy (**B** + **E**) stratified by synchronously-measured ctDNA% dichotomized by median (**A** + **B**) or by predefined bins (high, low, undetectable) (**D** + **E**)—see Supplementary Data 2 for per-patient ctDNA% values. Shading indicates 95% confidence intervals; in-set tables show univariable hazard ratios (HRs) from a Cox proportional hazards model. Forest plots show HRs and 95% confidence intervals from univariable (**C**) and multivariable (**F**) Cox proportional hazard regression models incorporating ctDNA% plus additional clinical prognostic markers. **G** Waterfall plot showing best PSA response (relative to baseline PSA) on first-line mCRPC therapy stratified by baseline ctDNA% (ctDNA > 30%, ctDNA 2-30%, and ctDNA < 2%). *P*-values (two-sided) reflect Fisher's Exact Test's comparing the proportion of patients achieving a ≥50% PSA response across ctDNA categories.

**H** Evidence for a nonlinear relationship between ctDNA% and risk of death. Univariable Cox proportional HRs (plotted as dots) for overall survival from initiation of first-line mCRPC therapy as a function of ctDNA% partitioned into non-overlapping intervals. Each interval is demarcated by the horizontal gray lines, with the center of each ctDNA% interval used as each datapoint's x-coordinate. Vertical gray lines show individual intervals' 95% HR confidence intervals. For all comparisons, the reference group is patients with ctDNA < 2%; marker size is proportional to the number of patients in the non-reference group (per-interval n is provided in the Source Data file). Solid red line shows a three-parameter negative exponential (with upper asymptote) curve fit. See Supplementary Data 4–6 for a complete summary of univariable and multivariable Cox proportional hazard regression model statistics, per-endpoint event rates, and summary of missing clinical data per initiating line of therapy. Correction for multiple hypothesis testing was not performed. REF reference.

Importantly, ctDNA% predicted outcomes as both a continuous variable and irrespective of ctDNA% dichotomization approach[14] (Fig. 4A–E). Future optimization of ctDNA%-based prognostication should explore alternative risk groups and/or opportunities for tailoring to specific clinical scenarios. Dichotomizing patients into non-overlapping ctDNA% prognostic groups enables convenient and easily-actionable stratification, but analogous to most existing clinical prognostic factors, risks oversimplifying the relationship

between ctDNA% and outcomes (Fig. 4G). However, non-arbitrary thresholds are also challenging to derive, and binning a continuous variable may cause false positive/negative patient allocations. We believe that the ctDNA% risk categories validated herein (low, medium, and high) provide a useful working model for the immediate clinical implementation of ctDNA% as a prognostic aide in mCRPC. cfDNA concentration was also prognostic for outcomes (Supplementary Fig. 8; Supplementary Data 4), although the effect

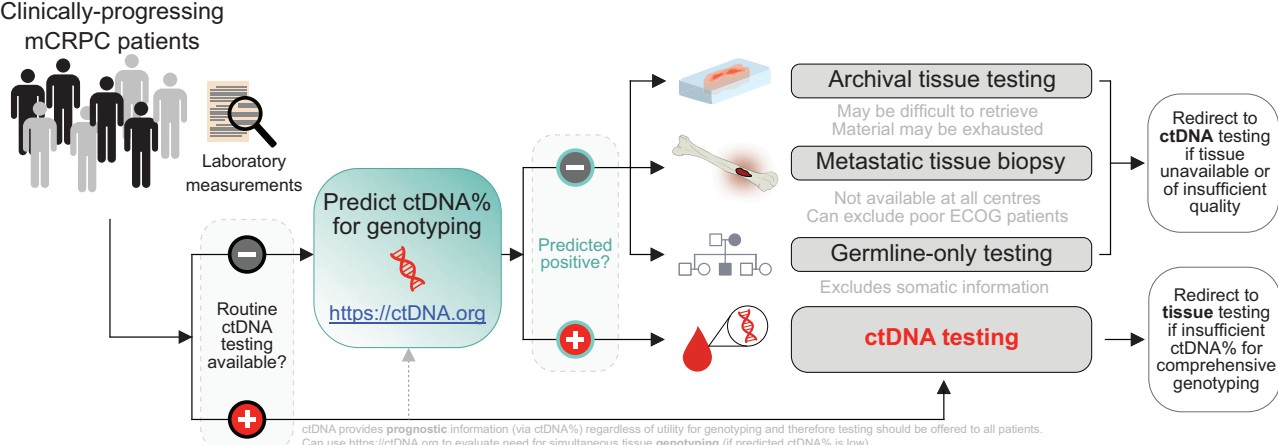

**Fig. 5 | Workflow for optimized clinical biomarker profiling for mCRPC incorporating ctDNA testing.** The ctDNA prediction tool (available at http://ctDNA.org) can be used to inform optimal strategy for mCRPC genomic biomarker testing, offering guidance as to whether to pursue ctDNA or tissue-based genotyping (in resource-limited circumstances where both testing modalities cannot be pursued simultaneously). Tissue-based genotyping should be initiated for patients with low predicted ctDNA%. However, ctDNA testing also offers valuable prognostic information (via ctDNA%) regardless of ctDNA%-sufficiency for sensitive genotyping, and therefore should be offered to patients if available as a prognostic adjunct (potentially in tandem with tissue-based genotyping). The tool's output includes the probability of a sample having ctDNA ≥ 2% and a point estimate of predicted plasma ctDNA%. Finally, the ctDNA%-prediction tool is flexible to any combination of missing data as well as differences in laboratory reference range values for lactate dehydrogenase and alkaline phosphatase.

size (e.g., hazard ratio) was consistently smaller than ctDNA%. In theory, the weaker prognostic stratification via cfDNA concentration may be pragmatically offset by the relative expediency and lower cost of quantification (compared to ctDNA%); however, ctDNA sequencing has the notable advantage of enabling simultaneous prognostication (via ctDNA%) and analysis of prognostic and treatment-predictive genomic alterations. Integrating ctDNA% with somatic information (e.g., *TP53* defects[20]) may provide additional prognostic granularity albeit potentially at the cost of added implementational complexity. Unfortunately, optimal ctDNA%-thresholds for risk stratification will likely differ by cancer type, meaning that our derived thresholds for mCRPC should not be extrapolated in a tumor-agnostic manner[52].

Patients with low ctDNA had exceptionally good prognosis but our data do not inform whether further stratification of this <2% subgroup would be helpful. Tumor-informed cfDNA assays originally developed for minimal residual disease can permit detection of ctDNA below 0.001%, given sufficient cfDNA input[41,53–55]. While unproven, prior data from primary cancers suggest that all patients with radiographic metastases will have detectable ctDNA with sensitive tumor-informed approaches[36,39,41,56]. However, these ultrasensitive assays would add operational complexity (i.e., archival tissue block retrieval and cancer genotyping) and exclude patients for whom tumor tissue is unavailable; ultimately it is plausible that more granular stratification of the ctDNA < 2% subgroup may offer diminishing returns for prognostication in advanced disease where ctDNA is relatively abundant. Conversely, measuring ctDNA% in the range of 2–100% is readily achieved via a variety of existing panel-based de novo genotyping strategies[23,29,57,58], including assays that have been deployed across recent phase II-III mCRPC trials[16,30,59–61]. In future, continued improvements in methodological transparency and acknowledgement of test limitations will be important for truly optimized ctDNA% testing, and ultimately end-users should still be aware of factors such as clonal-hematopoiesis and tumor aneuploidy that can cause erroneous ctDNA% readouts[26,33]. Newer generations of research mutation-based and methylation assays may be capable of comparably ultrasensitive ctDNA% detection without needing tissue[53–55,62], but are largely untested in prostate cancer and are not currently available to clinicians. Ultimately, it is plausible that further prognostic stratification of the ctDNA < 2% subgroup will provide diminishing returns in an elderly population (with generally good cancer-specific prognosis) where life expectancy is increasingly dictated by non-cancer comorbidities.

Same-patient ctDNA% was highly stable across serial progressions. This suggests limited utility of repeat sampling in the event of initially low ctDNA% (i.e., that would prohibit sensitive detection of all biomarker classes), and patients with ctDNA < 2% at progression should therefore be prioritized for biomarker analysis of archival tissue or metastatic biopsy (Fig. 1I, J). Likelihood of conversion to >30% ctDNA in a subsequent sample was especially low, occurring in only 8.6% of patients. In current clinically-validated biomarker tests, 30% ctDNA is the approximate limit of detection for deletions which constitute a major class of genomic eligibility for targeted drugs in prostate cancer (e.g., *BRCA2* and PARP inhibitors, *PTEN* and AKT/PI3K inhibitors)[23,61,63]. While more sensitive tumor-agnostic ctDNA sequencing approaches can largely overcome low ctDNA% for de novo detection of single nucleotide polymorphisms and indels[41,53,54], sensitivity for focal deletions cannot generally be improved beyond a fundamental lower limit of ~5-15% ctDNA[26,58]. Although the interval between sample requisition and results reporting is typically longer for tissue than ctDNA[40,64], potential delays are more tolerable for patients with excellent prognosis (i.e., ctDNA < 2%) where urgency of biomarker-informed clinical management is ostensibly lower. Nevertheless, our data indicates that serial ctDNA% closely mirrors clinical metrics of tumor aggression, meaning that patients with initially low ctDNA% who subsequently experience rapid clinical deterioration (and/or strong shifts in markers linked to ctDNA%, e.g., LDH) may be suitable candidates for ctDNA biomarker re-testing (Fig. 1K; Fig. 5). Generalizability of our ctDNA%-prediction tool to second- and third-line mCRPC may guide application of serial ctDNA genotyping, helping facilitate detection of resistance mechanisms or reevaluate eligibility for precision oncology trials or targeted treatments (Fig. 3E).

Biomarker testing in advanced cancers is highly time-sensitive, meaning that futile tests can deprive patients of biomarker-informed clinical management. Advantages of ctDNA profiling include its capacity to capture global metastatic biology (in contrast to single-core tissue biopsy) while simultaneously providing prognostic information via ctDNA%[65]. However, ctDNA profiling is only informative for somatic alterations if ctDNA% is relatively high, creating a dilemma over

optimal biomarker testing strategy. We developed a publically available web tool to guide users on whether to pursue blood collection for ctDNA genotyping versus prioritization of archival tissue retrieval or re-biopsy. The output generated from this machine-learning tool (i.e., probability that ctDNA-based genotyping is likely to be informative for somatic biomarkers in a patient with mCRPC) is based on consensus detection thresholds for mutations in clinically-relevant genes (e.g., *BRCA2*) in contemporary commercial tests[23,34]. Additionally, our tool can facilitate selective ctDNA genotyping in patients where tissue acquisition is not possible, and financial or logistical constraints prohibit routine indiscriminate ctDNA testing. For improved accuracy, the model can additionally utilize plasma cfDNA concentration which is inexpensive and expedient to measure, with extraction from blood plus quantification taking only a few hours. Since ctDNA% is being explored in other cancers as a prognostic tool[13,14,40,66], we anticipate that our results will serve as a blueprint of similar predictive tools across the spectrum of metastatic malignancies.

Our study has several limitations. First, our ctDNA% estimation approach did not account for whole-genome doubling events, which are challenging to infer from narrow targeted sequencing and can cause overestimation of ctDNA% due to inflated somatic allele frequencies. Second, we did not record the time of day of plasma cfDNA collection, which could plausibly affect ctDNA% via prior physical activity and hypothesized circadian variation in cfDNA release[67–69]. Third, the left-censored and right-skewed distribution of ctDNA% in mCRPC poses a challenge for supervised learning approaches that aim to predict ctDNA% (Fig. 1F)—this unbalanced dataset (with limited availability of high ctDNA samples for training) may potentially contribute to suboptimal classification accuracy. Interpretation of our ctDNA%-prediction tool's output may become more challenging for future generations of increasingly sensitive genotyping assays (with limits of detection <<0.5–1% VAF). Fourth, our study contained relatively few patients receiving first- or second-line taxane chemotherapy, with most chemotherapy-treated patients sourced from a single clinical trial enriched for poor prognosis features[16]. Small numbers and risk of selection bias precluded examination of potential interactions between treatment class (e.g., ARPI versus taxane) and ctDNA%. It is plausible that differences in prior treatment exposure may modulate tumor-intrinsic or -extrinsic determinants of ctDNA% at future timepoints, as well as the effect size of ctDNA% for prognosticating subsequent lines of therapy. Furthermore, ctDNA% may have subtly varying prognostic significance for different classes of subsequent treatment (i.e., is a predictive biomarker). Analysis of large clinically-standardized randomized cohorts will be required to uncover potential interactions between drug class (and/or mechanism of action) and ctDNA%. Importantly, the prognostic or predictive implications of ctDNA% remain largely undefined in the context of recent additions to the mCRPC therapeutic armamentarium (e.g., PARP inhibitors and Lutetium-177-PSMA-617 radioligand therapy). Finally, although we did not collect self-reported race or other measures of patient genetic background, based on the demographics of the jurisdictions contributing to our metacohort and validation cohorts we can assume that patients were primarily of European ancestry. Considering that ancestry may potentially interact with the relationships of ctDNA% to clinical/radiographic features and outcomes, caution should be exercised when extrapolating these correlations to diverse populations.

## Methods

### Cohort and clinical endpoints

We studied patients from: (1) two published randomized multicentre phase II trials addressing treatment involving standard-of-care drugs for first- and second-line mCRPC with a crossover design at progression (NCT02125357 and NCT02254785)[16,30]; and (2) a prospective province-wide plasma cfDNA biobanking program at the Vancouver Prostate Centre and BC Cancer[70] (Supplementary Fig. 1). All samples

were collected between October 2014 and October 2020, and time of last follow-up (for longitudinal clinical outcomes) was frozen in October 2021. All patients had histologically-confirmed prostatic adenocarcinoma (high-grade neuroendocrine and/or small-cell components were permitted), radiographic evidence of metastatic disease by conventional imaging (CT or bone scintigraphy), and castration-resistant prostate cancer defined as biochemical (PSA) or imaging progression despite castration levels of testosterone (PCWG3 criteria)[10]. Patients with active concurrent malignancy were excluded. Chemotherapy and/or AR targeted therapy administered in the castration-sensitive setting was permitted. Sex and/or gender are not relevant for any findings in this study and were therefore not incorporated into study design, clinical data collection, nor execution of specific analyses. Prostate cancer only affects people born as biological males, and our cohort includes people with aggressive prostate cancer irrespective of gender identity. All samples are de-identified at time of collection, and all researchers are blind to patient gender identity and gender presentation.

Plasma cfDNA samples must have been collected within 31 days prior to initiation of first, second, or third-line mCRPC treatment but not during active concurrent treatment[18,50,71,72]. This collection interval was selected (1) to minimize the confounder of treatment-induced ctDNA% suppression, since ctDNA abundance rapidly declines after the initiation of effective treatment for metastatic disease but typically recovers at time of progression; (2) to ensure the relevancy of our findings to a highly clinically-significant decision point, where patients are terminating prior therapy and being clinically re-evaluated to determine next line of therapy; (3) such that timing of matched cfDNA and routine clinical variables was approximately time-matched and standardized between patients. This 31-day period achieved an appropriate balance between facilitating considerations #2 and #3 while maintaining broad inclusivity to patients in our real-world ctDNA biobank (where ctDNA collections are more variably timed compared to the clinical trial cohorts). All patients must have provided at least a first-line mCRPC sample (Supplementary Fig. 1). An exception was made for patients enrolled in NCT02254785 who were permitted to receive one prior course of docetaxel for treatment-naive mCRPC prior to trial enrollment and two patients enrolled in NCT02125357 who did not have a cfDNA sample associated with first line treatment. These patients only provided cfDNA samples associated with second- and/or third-line treatment.

Audited clinical characteristics and outcomes data are published for the two randomized clinical trials (NCT02125357 and NCT02254785)[16,21,30]. For NCT02125357, wherever patients consented, medical records were manually reviewed for clinical data associated with subsequent post-protocol lines of therapy (i.e., not collected as part of the original trial). For the provincial cfDNA biobank program, clinical data was retrieved from manual review of electronic medical records. Clinical data for all three cohorts included patient demographics, clinical, pathological and laboratory features at the time of prostate cancer diagnosis and prior to each documented line of therapy, as well as time-to-event outcomes (Supplementary Data 1). Clinical endpoints evaluated in this study were OS, prostate-specific antigen (PSA) progression-free survival (PFS) and PSA response rate. PSA response was defined as ≥50% PSA decline from baseline pretreatment measurement, calculated using the on-treatment PSA nadir (standard PCWG2 criteria). PSA-PFS (on first, second or third-line mCRPC therapy) was defined as the time from start of therapy to PSA progression or death. PSA progression was defined as an increase of at least 2 μg/L and ≥25% from nadir. For patients with no PSA decline, PSA progression was defined as an increase of ≥2 μg/L and ≥25% from baseline. Calculation of PSA progression did not require a subsequent confirmatory PSA measurement collected 2 weeks following initial PSA rise, although if a subsequent appropriate measurement was available, an additional PSA increase was required in order to meet progression.

OS was defined as time from therapy initiation to time of death from any cause or last follow-up.

Approval for collection and profiling of patient samples was granted by the University of British Columbia Research Ethics Board (certificate numbers H18-00944, H16-00934). The study was conducted in accordance with the Declaration of Helsinki, and written informed consent was obtained from all patients prior to enrollment. Patients were not compensated for their participation in our study.

**Blood sample processing, library preparation, and sequencing**
Blood samples collected from 292 patients who participated in two completed randomized multicenter phase II trials (NCT02125357 and NCT02254785) were processed and sequenced as previously described[16,20,21]. New blood samples from 199 patients enrolled in the prospective province-wide cfDNA biobanking program were processed using the same protocols as NCT02125357 and NCT02254785. Briefly, whole blood was collected in either $4 \times 6$ ml EDTA tubes (2014–2016; BD, USA) or $2 \times 9$ mL Streck Cell-Free DNA BCT tubes (2016–2022; Streck, USA). Samples collected in EDTA tubes were centrifuged at $1600 \times g$ for $2 \times 10$ min at $4\,°C$ within 1–2 h of collection, and the plasma and buffy coat fractions were separated and stored at $-80\,°C$ prior to DNA extraction. Samples collected in Streck tubes were maintained at room temperature prior to and during processing. Streck tubes were centrifuged at $1600 \times g$ for 15 min, buffy coat was aliquoted, and plasma was transferred to a new tube and spun for ten additional minutes at $5000 \times g$ (or maximum attainable speed). Patient matched buffy coat and plasma were obtained simultaneously and stored at $-80$ prior to DNA extraction.

CfDNA was extracted from up to 6 mL of plasma with the Qiagen (Germany) Circulating Nucleic Acids kit, and quantified using the Quantus Fluorometer and QuantiFluor ONE dsDNA system (Promega, USA) or Qubit 2.0 Fluorometer and Qubit dsDNA HS Assay Kit (Thermo Fisher Scientific, USA). Matched germline DNA (gDNA) was extracted from the buffy coat fraction using the Qiagen (Germany) DNeasy Blood and Tissue Kit, or the Maxwell® RSC Blood DNA Kit and Maxwell® RSC Instrument (Promega, USA). Extracted gDNA was quantified with a NanoDrop spectrophotometer.

Targeted DNA sequencing of all cfDNA and gDNA samples was carried out using custom Roche NimbleGen SeqCap EZ Choice capture panels (Roche, Switzerland) comprised of ~72 mCRPC genes, as previously described[16,21,50,70]. Final enriched library pools were sequenced on Illumina MiSeq ($2 \times 300$ bp), NextSeq ($2 \times 150$ bp), or HiSeq 2500 ($2 \times 125$ bp) instruments. Sequence alignment and somatic variant calling was performed using an established validated pipeline as previously described[65,70].

**Estimation of ctDNA fraction**
For all samples in our metacohort, we utilized a standardized hierarchical approach for estimating ctDNA fraction (ctDNA%) predicated on (1) somatic mutation variant allele frequencies (corrected for statistical outliers and potential concurrent loss-of-heterozygosity; LOH), and (2) germline heterozygous single nucleotide polymorphism allele frequency (HSAF) deviation from 50% heterozygosity in genes with evidence of LOH. Our established published approach for estimating ctDNA% from targeted panels shows high analytical concordance to ctDNA% estimates from whole-exome (via copy number model fitting and somatic mutation allele frequencies)[20,21] and deep whole-genome sequencing data (via copy number model fitting using a bespoke pipeline [code available; see Data availability], as well as application of accepted tools ASCAT and Battenberg)[65,73,74]. Prior validation includes both whole-exome and deep whole-genome sequencing of a subset of cfDNA samples also analyzed herein (i.e., samples procured from NCT02125357).

Specifically, mutation-based ctDNA% was calculated from the variant allele fractions (VAF) of autosomal somatic mutations on non-amplified genes (log-ratio <0.2). Since mutant allele fraction is elevated during concurrent loss of the other wildtype allele (i.e., LOH) which may not be possible to detect when ctDNA% is low, we conservatively assumed that all somatic mutations could be associated with LOH. In regions of LOH, mutation VAF and ctDNA% (both as variables with lower and upper bounds of 0 and 1, respectively) are mathematically related as $\text{ctDNA}\% = 2/(\text{VAF}^{-1} + 1)$. To account for sampling noise, we modeled the mutant read count as arising from a binomial distribution, and conservatively calculated what the true VAF would be if the highest observed VAF was a 95% quantile outlier[21]. After calculating a ctDNA% estimate for each somatic mutation, the highest estimate was adopted as the overall estimate for the sample under the assumption that this mutation was the most likely to be truncal to the metastatic lineage. Germline variants, stereotypical sequencing and alignment artifacts, and clonal-hematopoiesis of indeterminate potential can confound somatic mutation-based estimation of ctDNA% (i.e., masquerade as tumor-derived variants resulting in false-positive estimates)[26,33]. Importantly, these potential confounders are largely eliminated through our parallel deep sequencing of patient-matched white blood cells and paired variant calling strategy.

For internal validation of ctDNA%, we also applied an orthogonal copy number-based approach for measuring ctDNA% leveraging HSAF deviation from heterozygosity in genes with LOH. Germline SNPs were identified from paired normal white blood cell samples as any variant present in the ExAC, Kaviar, or gnomAD databases with sufficient coverage. We determined all heterozygous intragenic SNPs located on genes with evidence for a single-copy deletion (log-ratio between $-0.3$ and $-0.7$). Genes were excluded from ctDNA% calculation if they contained <4 unique SNPs. We calculated the median major allele frequency (i.e., $|0.5 - \text{VAF}| + 0.5$) of SNPs within each gene and propagated this value through $\text{ctDNA}\% = 2 - \text{VAF}^{-1}$. Because copy number-based approaches for estimating ctDNA% are not ideally suited for narrow targeted panels, we defaulted to ctDNA% values produced by the mutation-based approach except in three samples that lacked eligible mutations (but contained copy number evidence of quantifiable ctDNA%). Finally, there were 27 samples where both mutation- or copy number-based ctDNA% estimates were uninformative (mainly false-positive or -negative ctDNA% estimates based on inconsistent and/or low-quality evidence for prostate cancer derived ctDNA). In these samples, we provided a conservative qualitative estimate of ctDNA% based on the average log-ratios of genes harboring putative heterozygous deletions or low-level copy gains. In 6 samples, no eligible mutations or copy number changes were detected except for an isolated *AR* gain, and the ctDNA% of these samples was heuristically set at 5%. The detection of an *AR* gain in this context rules out the possibility of ctDNA < 2%, and we selected 5% ctDNA since this represents the approximate minimum ctDNA% limit of detection for a ~7–8 copy *AR* gain (i.e., the average *AR* copy number in mCRPC samples with evidence of an *AR* gain from log-ratio evaluation[20]). All somatic variants, sample copy number profiles, and ctDNA% estimates were manually evaluated (mutations inspected using Integrated Genomics Viewer 2.12.3)[75].

ctDNA% prognostic risk categories of high (30–100%), low (2–30%), and undetectable (<2%) were predefined[16,30]. Category thresholds were heuristically selected to (1) achieve an approximately balanced dichotomization of patients commencing first-line therapy for mCRPC (within the range of quantifiable ctDNA% as dictated by our methodology; i.e., 2–100% ctDNA), and (2) mirror the typical ctDNA limit of detection for single-copy deletions (30%) in current commercially-available genotyping assays (i.e., so that the threshold of 30% is simultaneously meaningful for biomarker evaluation).

**Machine-learning prediction models**
We leveraged two established supervised learning techniques to predict ctDNA% from clinical variables: gradient boosting

(XGBoost version 1.7.0) and K-nearest-neighbors (KNN) (for orthogonal validation of our primary gradient boosting framework). Both sets of models were trained on the $n = 463$ baseline mCRPC patients (with time-matched ctDNA%). Although we collected data from subsequent lines of therapy that theoretically could have been used for model training and evaluation (to augment our cohort size), we restricted training to baseline samples to ensure all measurements reflect unique patients (i.e., such that the dataset is independent and identically-distributed).

The XGBoost models were trained using various subsets of 17 clinical variables including diagnostic variables, pre-mCRPC clinical outcomes, and laboratory and radiographic variables measured as initiation of first-line mCRPC therapy (Supplementary Data 6). Feature pre-selection was not performed for the XGBoost models, instead allowing XGBoost to independently infer the importance of each variable. Known markers of tumor burden (e.g., PSA, LDH, ALP, and cfDNA concentration) were constrained to a monotonic relationship with ctDNA% (predicated on our biological observations in Fig. 2) after testing that these constraints did not negatively impact the model's accuracy. Other clinical variables with a less clear relationship with ctDNA% (e.g., patient age) were not monotonically constrained. No feature normalization techniques were applied (given that XGBoost is invariant to feature scaling). Missing values in training and validation sets were separately imputed using KNN imputation (with $K = 3$ after scaling all dimensions to have a standard deviation of 1). Simple median imputation was rejected because it biases samples with missing data towards the cohort average. XGBoost's built-in sparsity-aware split finding method for handling missing data was similarly rejected because its results are unpredictable in situations where an input variable is dense in the training data, but sparse during model use (a common scenario for our website implementation of this model). Furthermore, XGBoost's built-in missing data handling assumes that the underlying causes and correlates of missing data are the same in the training data as when the model is used, which is not necessarily true.

The XGBoost models were optimized for binary logistic classification of ctDNA <2% or ≥2%. This classification threshold is based on the approximate ctDNA% limit of detection for common commercial and laboratory-based ctDNA genotyping assays which are widely used in prostate cancer. Our targeted ctDNA assay has an approximate conservative lower limit of detection for somatic mutations of ~0.5–1% VAF, corresponding to ~1-2% ctDNA% (assuming the mutation is clonal), mirroring the performance characteristics of established ctDNA companion diagnostic tests. Model training, hyperparameter tuning, and generalized accuracy evaluation was performed via nested cross-validation (i.e., K*L-fold cross-validation). Briefly, we split the $n = 463$ patients into training and validation sets using 20-fold cross-validation. The outer training dataset was used for hyperparameter tuning by subdividing it into inner training and validation sets using 5-fold cross-validation. We employed an adaptive grid search with `num_round` ∈ [10, 20, 50, 100, 200, 500], `max_depth` ∈ [2, 3, 4, 5, 6, 7, 8], eta ∈ [0.005, 0.01, 0.02, 0.05, 0.1], and subsample ∈ [0.25, 0.5, 0.75] using identical training-validation set splits for every tested hyperparameter combination, thereby ensuring that the performance of each hyperparameter set was not affected by unnecessary random sampling. After each iteration of the adaptive grid search, we eliminated the 50% of hyperparameter candidates that achieved the lowest accuracy (quantified via AUC), and in the next iteration doubled the number of cross-validation rounds for the remaining hyperparameter candidates, thereby ranking the hyperparameters by accuracy with increasing precision. Hyperparameters `colsample_bytree` and `colsample_bylevel` were fixed to the same value as `subsample`, to manage the size of the hyperparameter space.

To interpret how individual clinical variables affect the 17-feature XGBoost model's predicted probabilities, we used the Python package `shap` 0.41.0 to calculate SHAP (SHapley Additive exPlanation) scores, which quantify how much a clinical variable's value in a given sample impacted the predicted probability[76]. The scores were visualized as a barplot showing the average SHAP score of a specific clinical variable category or quartile across all samples (Fig. 3C).

For our ctDNA%-prediction web tool available at https://www.ctDNA.org, we selected 8 clinical variables with high SHAP scores (Fig. 3C) and anticipated generalizability across clinical contexts, and trained a separate XGBoost classification model for every possible combination of missing input data (i.e., $n = 255$ combinations). This strategy for handling missing input data is more robust to scenarios where users provide highly incomplete data (e.g., only a single clinical variable), since in such scenarios the predictions would be almost entirely based on the imputation algorithm and the proven accuracy of gradient boosting would be lost. These models were trained on all 738 samples in the cohort, using the automated training procedure described in the previous paragraph. For the final full 8-variable model, the automated hyperparameter tuning procedure selected the parameters `num_round = 500`, `max_depth = 5`, `eta = 0.02`, `subsample = 0.25`, `colsample_bytree = 0.25`, and `colsample_bylevel=0.25`.

We also trained an XGBoost regression model optimized for predicting sample ctDNA fraction as a continuous variable for use on our website (the classification and regression results are both shown to the user after inputting a patient's clinical data). This model was trained using the same cohort and automated procedure, but was optimized for mean absolute error. For the final full 8-variable model, the hyperparameters `num_round=200`, `max_depth = 2`, `eta = 0.005`, `subsample = 0.75`, `colsample_bytree = 0.75`, and `colsample_bylevel = 0.75` were selected by the automated procedure.

To validate that our gradient boosting prediction model outperforms simpler models, we also trained a dimensionally-weighted K-nearest neighbor model (with $K = 20$) for predicting ctDNA% (as a continuous variable) and for estimating the probability that ctDNA>2% in a given sample. To predict ctDNA%, we calculated the median ctDNA% of 20 nearest neighbors. To predict P(ctDNA>2%), we calculated the fraction of 20 nearest neighbors with ctDNA > 2%. Dimensional weights minimizing mean absolute error between predicted and true ctDNA% for the KNN model were learned via Nelder-Mead optimization using the training set. Accuracy of the KNN model was evaluated using leave-one-out cross-validation. Missing input variables were simply omitted when calculating dimensionally-weighted Euclidean distances to find nearest neighbors.

External validation of the parsimonious 8-variable XGBoost ctDNA% prediction model was performed using two external clinical trial datasets with documented sequencing-based ctDNA% estimates. For both cohorts, plasma cfDNA was collected prospectively prior to mCRPC treatment initiation and time-matched to clinical data (measured and assembled independently by the respective trial investigators). OPT/ILU cohort: 84 plasma cfDNA samples collected from 84 patients with mCRPC across two prospective multi-center observational studies in the Netherlands (NCT02426333 [OPTIMUM]; NCT02471469 [ILUMINATE]). Patients were initiating first-line treatment with abiraterone acetate plus prednisolone or enzalutamide[50]. ProBio cohort: 307 plasma cfDNA samples collected from 307 patients with mCRPC at screening to determine eligibility for the multi-center, multi-arm biomarker-driven ProBio platform trial open in four European countries (NCT03903835 [ProBio])[77,78]. Screened samples used for validation include those with undetectable ctDNA (even though patients with ctDNA-negative samples are excluded from subsequent trial enrollment). All plasma samples were collected, processed, sequenced, and bioinformatically analyzed by the ProBio investigators

using different preanalytical and bioinformatic methodology than what was used for our original metacohort.

## Statistical analyses and reproducibility

Statistical tests and data analyses were conducted in Python 3.7 (using pandas 0.25.0, numpy 1.16.4, scipy, and statsmodels). Visualizations were generated using matplotlib (Python). All boxplots are centered at the median and display interquartile ranges (IQR) and minima and maxima extending to 1.5× IQR (per convention). Human Figure (Fig. 1A) was obtained and modified from Wikimedia Commons (original authors: Patrick J. Lynch (medical illustrator); C. Carl Jaffe, MD (cardiologist); Yale University Center for Advanced Instructional Media) available under a Creative Commons Attribution 2.5 generic license (https://commons.wikimedia.org/wiki/File:Skeleton_whole_body.svg). Descriptive statistics were used for post hoc exploration of baseline clinical characteristics and ctDNA%. Survival functions for time-to-event outcomes (e.g., PSA-PFS and OS) were estimated using the Kaplan-Meier method. Hazard ratios were calculated using univariable and multivariable Cox proportional hazards models via lifelines v0.26.4, and p-values reflect the Wald test of a single parameter. All hypothesis tests were two tailed and required a 5% significance threshold. Correction for multiple hypothesis testing was not performed. Sample size and a statistical analysis plan (including power calculations) was not formally prespecified for this retrospective exploratory study. Imputation for missing data for traditional statistical analyses was not performed (due to the high overall completeness of our data (Fig. 1B); patients with missing data were excluded from relevant descriptive analyses). For all multivariable Cox proportional hazards models, patients with ≥1 covariate with missing data are omitted. The statistical framework for evaluating the relationships between ctDNA%, clinical variables, and survival outcomes is outlined in Supplementary Fig. 9. This retrospective meta-analysis did not directly incorporate randomization, although the constituent clinical trials (NCT02125357 and NCT02254785) from which patients were accrued did involve random treatment assignment. Random permutation sampling was performed in the context of nested cross-validation for developing our ctDNA-fraction prediction tool. Investigators were not blinded to any patient data, patient allocation during experiments, or outcomes assessment.

## Reporting summary

Further information on research design is available in the Nature Portfolio Reporting Summary linked to this article.

## Data availability

The human reference genome hg38 was downloaded from UCSC. Germline variant population frequency is available at gnomAD v3.0 (https://gnomad.broadinstitute.org/). De-identified sequencing data from patients included in this study have been deposited to the European Genome-Phenome Archive (EGA) database under the accession code EGAS50000000211 (available at: https://ega-archive.org/studies/EGAS50000000211). Sequencing data are available indefinitely for research use only under standard EGA controlled access: data access inquiries should be directed to A.W.W. (awwyatt@mail.ubc.ca). Timeframe for data access will be subject to EGA policy and process. All other data supporting the findings of this study are available within the article (including its Supplementary Data and Source Data files). Source data are provided with this paper.

## Code availability

Custom computer code utilized for our machine-learning models is available on GitHub at https://github.com/annalam/ctdna-prediction-manuscript. Our complete ctDNA somatic variant calling pipeline is also available on GitHub (https://github.com/annalam/cfdna-wgs-manuscript-code) and is described in detail in a prior publication[65].

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

## Acknowledgements

This work was primarily funded by a Canadian Cancer Society Challenge Grant (grant #707339) to A.W.W., K.N.C., and N.M.F. Other funding support was provided by the Canadian Institutes of Health Research, Prostate Cancer Canada, Movember Foundation, Prostate Cancer Foundation, Jane and Aatos Erkko Foundation, Academy of Finland Center of Excellence program (project no. 312043), a Terry Fox New Frontiers Program Project Grant, the BC Cancer Foundation, and Kom op tegen Kanker (Stand Up To Cancer - Flemish Cancer Society) [grant numbers STI.VLK.2020.0006.01; STI.VLK.2022.0005.01]. No funding sources were involved in the design or execution of the study. The authors are grateful to all participating patients and their families.

## Author contributions

N.M.F.: Conceptualization, data curation, software construction, formal analysis, visualization, methodology, writing-original draft, project administration, writing-review and editing; C.M.D: Conceptualization, data curation, formal analysis, methodology, writing-original draft, project administration, writing-review and editing; C.H.: Conceptualization, data curation, software construction, formal analysis, visualization, methodology, writing-original draft, project administration, writing-review and editing; W.T.: Software construction, formal analysis, visualization, methodology, writing-original draft, writing-review and editing; W.F.: Data curation, writing-review and editing; A.J.M.: Software construction, formal analysis, visualization, writing-review and editing; Catarina.K.: Data curation, writing-review and editing; E.M.K.: Data curation; formal analysis, methodology, writing-review and editing; K.P.: Data curation, validation, methodology, writing-review and editing; E.S.: Data curation, validation, methodology, writing-review and editing; C.Q.B.: Data curation, validation, methodology, writing-review and editing; G.D.: Data curation, validation, methodology, writing-review and editing; S.W.S.N.: Data curation, validation, methodology, writing-review and editing; T.S.: Data curation, validation, methodology, writing-review and editing; J.V.: Resources, writing-review and editing; K.N.: Resources, writing-review and editing; D.L.F.: Resources, writing-review and editing; M.Z.: Resources, writing-review and editing; S.M.: Resources, writing-review and editing; S.P.: Resources, writing-review and editing; J.L.: Resources, writing-review and editing; E.H.: Resources, writing-review and editing; M.S.: Resources, writing-review and editing; L.N.: Resources, writing-review and editing; B.J.E.: Resources, writing-review and editing; Christian.K.: Resources, writing-review and editing; S.T.: Resources, formal analysis, methodology, writing-review and editing; M.N.: Resources, writing-review and editing; S.H.T.: Data curation, validation, methodology, writing-review and editing; E.B.: Data curation, validation, methodology, writing-review and editing; N.M.: Resources, writing-review and editing; N.P.v.E.: Resources, writing-review and editing; B.D.L.: Resources, validation, formal analysis, writing-review and editing; J.L.: Resources, validation, formal analysis, writing-review and editing; H.G.: Resources, validation, writing-review and editing; D.J.K.: Conceptualization, data curation, formal analysis, methodology, writing-review and editing; M.A.: Conceptualization, data curation, software construction, formal analysis, visualization, methodology, writing-original draft, project administration, writing-review and editing, supervision; K.N.C.: Conceptualization, resources, project administration, supervision, funding acquisition, writing-review and editing; A.W.W.: Conceptualization, resources, project administration, supervision, funding acquisition, writing-original draft, writing-review and editing. All authors reviewed and approved the final version of the manuscript.

## Competing interests

C.M.D. reports Honoria from MSD, Bristol-Myers Squibb, Medison and Pfizer and consulting fees from Biomica LTD. E.M.K. has served in consulting or advisory roles in Astellas Pharma, Janssen, Ipsen and received honoraria from Janssen, Ipsen, Astellas Pharma, and Research Review. E.M.K. also reports research funding from Astellas Pharma (institutional) and AstraZeneca (institutional), and travel expense reimbursement from Astellas Pharma, Pfizer, Ipsen and Roche. K.N. has served on advisory boards for Abbvie, Astellas, AstraZeneca, Bayer, Bristol-Myers Squibb, Merck, Janssen and Tersera. D.L.F. has served in advisory roles or received honorarium from Janssen, Bayer, Astellas, AstraZeneca and Pfizer. M.A. is a shareholder in Fluivia Ltd. K.N.C. reports grants from Janssen, Astellas, and Sanofi during the conduct of the study. K.N.C. also reports grants and personal fees from Janssen, Astellas, AstraZeneca, and Sanofi, as well as personal fees from Constellation Pharmaceuticals, Daiichi Sankyo, Merck, Novartis, Pfizer, Point Biopharma, and Roche

outside the submitted work. A.W.W. has served on advisory boards and/or received honoraria from AstraZeneca, Astellas, Bayer, EMD Serono, Janssen, Merck, and Pfizer. A.W.W.'s laboratory has a contract research agreement with ESSA Pharma. The remaining authors declare no competing interests.

## Additional information

Nicolette M. Fonseca[1,18], Corinne Maurice-Dror[2,18], Cameron Herberts[1,18], Wilson Tu[1], William Fan[2], Andrew J. Murtha[1], Catarina Kollmannsberger[2], Edmond M. Kwan[1,2,3], Karan Parekh[1], Elena Schönlau[1], Cecily Q. Bernales[1], Gráinne Donnellan[1], Sarah W. S. Ng[1], Takayuki Sumiyoshi[1,4], Joanna Vergidis[5], Krista Noonan[6], Daygen L. Finch[7], Muhammad Zulfiqar[8], Stacy Miller[9], Sunil Parimi[2], Jean-Michel Lavoie[5], Edward Hardy[10], Maryam Soleimani[2], Lucia Nappi[1,2], Bernhard J. Eigl[2], Christian Kollmannsberger[2], Sinja Taavitsainen[11], Matti Nykter[11], Sofie H. Tolmeijer[1,12], Emmy Boerrigter[13], Niven Mehra[12], Nielka P. van Erp[13], Bram De Laere[14,15,16], Johan Lindberg[16], Henrik Grönberg[16], Daniel J. Khalaf[2], Matti Annala[1,11,19] ✉, Kim N. Chi[1,2,19] ✉ & Alexander W. Wyatt[1,17,19] ✉

[1]Vancouver Prostate Centre, Department of Urologic Sciences, University of British Columbia, Vancouver, BC, Canada. [2]Department of Medical Oncology, BC Cancer, Vancouver, BC, Canada. [3]Department of Medicine, School of Clinical Sciences; Monash University, Melbourne, VIC, Australia. [4]Department of Urology, Graduate School of Medicine, Kyoto University, Kyoto, Japan. [5]Department of Medical Oncology, BC Cancer, Victoria, BC, Canada. [6]Department of Medical Oncology, BC Cancer, Surrey, BC, Canada. [7]Department of Medical Oncology, BC Cancer, Kelowna, BC, Canada. [8]Department of Medical Oncology, BC Cancer, Abbotsford, BC, Canada. [9]Department of Radiation Oncology, BC Cancer, Prince George, BC, Canada. [10]Tom McMurtry & Peter Baerg Cancer Centre, Vernon Jubilee Hospital, Vernon, BC, Canada. [11]Prostate Cancer Research Center, Faculty of Medicine and Health Technology, Tampere University and Tays Cancer Center, Tampere, Finland. [12]Department of Medical Oncology, Research Institute for Medical Innovation, Radboud University, Nijmegen, The Netherlands. [13]Department of Pharmacy, Research Institute for Medical Innovation, Radboud University, Nijmegen, The Netherlands. [14]Department of Human Structure and Repair, Ghent University, Ghent, Belgium. [15]Cancer Research Institute Ghent (CRIG), Ghent University, Ghent, Belgium. [16]Department of Medical Epidemiology and Biostatistics, Karolinska Institute, Stockholm, Sweden. [17]Michael Smith Genome Sciences Centre, BC Cancer, Vancouver, BC, Canada. [18]These authors contributed equally: Nicolette M. Fonseca, Corinne Maurice-Dror, Cameron Herberts. [19]These authors have jointly supervised this work: Matti Annala, Kim N. Chi, Alexander W. Wyatt. ✉e-mail: matti.annala@tuni.fi; kchi@bccancer.bc.ca; awwyatt@mail.ubc.ca

