## [Peer Review File · Nature Communications]

Editorial Note: This manuscript has been previously reviewed at another journal that is not operating a transparent peer review scheme. The manuscript was considered suitable for publication without further review at Nature Communications.

REVIEWER COMMENTS

Reviewer #4 (Remarks to the Author):

In this study the authors explore the prognostic role of ctDNA in metastatic prostate cancer by analyzing a large cohort of patient samples and, more interestingly, develop a tool to predict ctDNA fraction based on several clinical parameters.

The work follows an elegant approach, with a uniform and harmonized analysis pipeline, which is well described in the methods section and supplementary Figures. The large cohort analyzed gives strength to the conclusions, however, it is important to recall that most of the samples used here had already been analyzed in previous publications from the same group (Annala et al, 2018, Cancer Discovery; Khalaf et al, 2018, Lancet Oncol.; Annala et al, 2021, Clin Cancer Res.) as now reported in Supplementary Figure 1. Furthermore, several works have already demonstrated the association between ctDNA fraction, clinical parameters associated with worse prognosis, and clinical outcome (Mehra et al, 2018, Eur Urol; Annala et al, 2018, Cancer Discovery; Vandekerkhove et al, 2019, Eur Urol 2019).

The strength of the study is that by performing a meta-analysis with a large cohort of patients (with previous reported cohorts and new patients), associating ctDNA with multiple clinical features, the authors delivered a powerful prediction tool to calculate ctDNA fraction, which might be used for biomarker studies and patient stratification in the clinics. The manuscript needs to extend its focus beyond the prognostic role of ctDNA, as this aspect lacks a significant conceptual advancement. Instead, it should center its narrative around the remarkable novelty that their AI tool potentially introduces to the clinical application of ctDNA testing. I have several comments/concerns.

Major comments:

- The prognostic association found for ctDNA is an important result due to the large cohort size presented here, however, the majority of the samples had already been used for this purpose in previous studies. I do not see the novelty of this part of the study beyond the metadata analysis and data analysis harmonization between multiple cohorts. In this sense, centering the title and the narrative of the work in the prognostic role of ctDNA in metastatic prostate cancer diminishes the importance of other novel aspects of the work, such as the predictive model developed in here.

- The authors acknowledge in the discussion that the effect of other therapies, such as taxane-based chemotherapy, could not be investigated in detail in this metacohort. However, from the data presented here it seems that patients exposed to Taxanes experienced an increase in both cfDNA concentration and ctDNA fraction (in Figure 1L mCSPC patients with taxane intensification have higher ctDNA; in Figure 1F-H patients in later lines of treatment tend to have more cfDNA and ctDNA); which makes sense since: 1) taxane-based chemotherapy induces apoptosis, which is the major source for cfDNA release and; 2) patients with more advanced disease, in later lines of treatment, are more frequently treated with

chemotherapy. Therefore, it will be worth to explore and add to the manuscript some figures (as the ones included in the NMed rebuttal letter) reflecting the effect that ARPI or Taxanes have in both cfDNA and ctDNA concentration and ctDNA fraction. How could this affect the prediction model?

As one would expect, cfDNA concentration and ctDNA fraction tends to be higher in 2L and 3L samples (Figure 1F-H), however, the authors show that the prediction model also works efficiently in this context. How does the model account for ctDNA variance due to different therapies or previous lines of treatment if these variables were not incorporated in the model?

- The authors demonstrate in several figures the tight association between cfDNA concentration and ctDNA fraction. In fact, they claim that cfDNA concentration seems to be the highest predictor of ctDNA fraction and cfDNA concentration is inter-correlated with other blood biomarkers such as LDH or ALP. However, they do not explore the association between cfDNA concentration and other clinical parameters (metastases, time to progression, etc.) or outcome. What would be the prognostic role of cfDNA concentration? How would it compare to ctDNA?

- Previous studies showed that the amount of input blood (ml of plasma) used for cfDNA isolation could affect cfDNA concentration yield (Alborelli et al., 2019, Cell Death Dis.). Could this also affect ctDNA fraction? If so, how could this affect the predictions of the model?

Minor comments:

- In Figure 1B, the number of patients with bone mets does not match the percentages shown in Figure 1D.

- In Figure 4G the y-axis legend is a bit confusing. Less ctDNA fraction is associated with a better PSA response (more patients with more than 50% PSA reduction), however, this is not completely clear with the representation shown in here.

Reviewer #5 (Remarks to the Author):

In this well-written manuscript, the authors provide a detailed analysis of the clinical determinants of circulating tumor DNA fraction (ctDNA%) and its utility for prognostication using a large metacohort of 491 mCRPC patients. Using this data, they then develop a machine learning-based tool that predicts the likelihood that a patient will have a sufficiently high ctDNA% for clinically informative ctDNA genotyping. The study highlights ctDNA% as a validated tool for patient risk stratification, and provides a potentially clinically useful web-based tool for estimating ctDNA% using clinical parameters, prior to actual ctDNA testing.

The authors have done an excellent job responding to the Reviewer comments. In particular, they are to be commended for performing validation studies of their ctDNA% prediction model with two additional external cohorts. The revised text also nicely clarifies the intended clinical application of their ctDNA% prediction machine learning-based tool.

I have the following additional minor comments:

Line 209; Fig. 3F – Can the authors speculate on the potential nature of the additional patient- or tumor-specific determinants of ctDNA% that are not included in their models?

Lines 239-241 – Please provide Tables showing the summary clinical characteristics of the two external prospective mCRPC datasets used for validation.

Line 266 – Please delete the extra period.

Line 279; Fig. 4A and 4D – The labeling of the X-axis is not clear and is not consistent with text in the Figure Legend. Should this be “Time from 1L therapy initiation to death” rather than “mCRPC diagnosis to death”?

Lines 426-429; Table 1 – Please add race as a clinical characteristic and provide a % breakdown of the racial distribution of patients within the metacohort and validation cohorts. The authors have already described the predominance of European ancestry patients as a limitation of their study, but it would still be useful to have the actual percentages presented in the Tables.

Reviewer #6 (Remarks to the Author):

The manuscript by Fonseca et al. describes the development of a novel tool for decision-making on whether cfDNA-based genotyping is viable based on the probability of ctDNA detection. In addition, they demonstrate the prognostic value of ctDNA levels in patients with advanced prostate cancer. The latter, while not entirely novel, it underscores previous work using a large meta-cohort, in which multiple confounders could be evaluated to demonstrate ctDNA% as an independent predictive biomarker.

Overall, the manuscript is very clear, and the study and analyses are well described.

A major comment relates to Figure 6, which summarises the potential clinical pipeline using the developed tool. However, from the figure, it is not apparent that the ctDNA prediction tool is the 2nd in the decision-making process. The fact that the arrow goes from patients to the tool, implies that it is the 1st step.

I just have some minor suggestions and queries:

1. While the changes to ‘ctDNA biomarker genotyping’ provide some clarity, it is not always appropriate. Such as, in this sentence in page 3: “Excitingly, ctDNA% is increasingly reported on commercially-available ctDNA biomarker genotyping tests 23, meaning that ctDNA%-prognostication is poised to rapidly influence patient management pending its clinical validation.”

2. Page 7: ‘credential’ is not a verb

3. Methods: There are no details on the supplier companies, country, etc, which is expected as per most journal guidelines.

4. Methods, page 17: The way blood processing is described, sounds like only Streck derived plasma samples were re-spun and used for buffy coat collection.

5. Methods, page 18, when referring to " $ctDNA\% = 2/(1/VAf + 1)$ " and also " $ctDNA\% = 2 - VAF-1$ ", I believe you are implying an adjustment factor rather than the %ctDNA itself. Please describe clearly. Otherwise, irrespective of VAF, all ctDNA values will fall below 2.

Response to Reviewers

Reviewer #4 (Remarks to the Author):

In this study the authors explore the prognostic role of ctDNA in metastatic prostate cancer by analyzing a large cohort of patient samples and, more interestingly, develop a tool to predict ctDNA fraction based on several clinical parameters.

The work follows an elegant approach, with a uniform and harmonized analysis pipeline, which is well described in the methods section and supplementary Figures. The large cohort analyzed gives strength to the conclusions, however, it is important to recall that most of the samples used here had already been analyzed in previous publications from the same group (Annala et al, 2018, Cancer Discovery; Khalaf et al, 2018, Lancet Oncol.; Annala et al, 2021, Clin Cancer Res.) as now reported in Supplementary Figure 1. Furthermore, several works have already demonstrated the association between ctDNA fraction, clinical parameters associated with worse prognosis, and clinical outcome (Mehra et al, 2018, Eur Urol; Annala et al, 2018, Cancer Discovery; Vandekerckhove et al, 2019, Eur Urol 2019).

The strength of the study is that by performing a metanalysis with a large cohort of patients (with previous reported cohorts and new patients), associating ctDNA with multiple clinical features, the authors delivered a powerful prediction tool to calculate ctDNA fraction, which might be used for biomarker studies and patient stratification in the clinics. The manuscript needs to extend its focus beyond the prognostic role of ctDNA, as this aspect lacks a significant conceptual advancement. Instead, it should center its narrative around the remarkable novelty that their AI tool potentially introduces to the clinical application of ctDNA testing. I have several comments/concerns.

Major comments:

- The prognostic association found for ctDNA is an important result due to the large cohort size presented here, however, the majority of the samples had already been used for this purpose in previous studies. I do not see the novelty of this part of the study beyond the metadata analysis and data analysis harmonization between multiple cohorts. In this sense, centering the title and the narrative of the work in the prognostic role of ctDNA in metastatic prostate cancer diminishes the importance of other novel aspects of the work, such as the predictive model developed in here.

Thank you for your thorough appraisal of our manuscript and helpful suggestions.

We recognize that 59% of patients in our study have been analyzed in prior clinical trial publications that touched in part on the prognostic implications of ctDNA%. However, it is important to emphasize that we have provided updated clinical outcomes for consenting trial patients (e.g. median 20.3 months f/u (range: 0.4-81.6) in our metacohort versus only 12.9 (0-32.1) in Annala et al. 2018), enhancing the statistical maturity of our ctDNA% outcomes analyses relative to these original trial publications. Our large metacohort has also enabled us to perform new analyses previously not possible due to smaller cohort size, such as searching for interaction effects between sets of risk-stratification variables, exploring non-linear relationships (e.g. between ctDNA% and risk of death), and looking across different lines of treatment.

To emphasize the novelty of our ctDNA%-prediction tool, we featured this result in our original Abstract and devoted a substantial portion of the Introduction to provide context for this advancement. In our revised submission, we have adjusted our manuscript title to more directly acknowledge the novelty of our ctDNA% prediction tool:

- ORIGINAL: “*Enhanced prognostication of advanced prostate cancer using ctDNA fraction*”
- NEW: “*Prediction of plasma ctDNA fraction and prognostic implications of liquid biopsy in advanced prostate cancer*”

- The authors acknowledge in the discussion that the effect of other therapies, such as taxane-based chemotherapy, could not be investigated in detail in this metacohort. However, from the data presented here it seems that patients exposed to Taxanes experienced an increase in both cfDNA concentration and ctDNA fraction (in Figure 1L mCSPC patients with taxane intensification have higher ctDNA; in Figure 1F-H patients in later lines of treatment tend to have more cfDNA and ctDNA); which makes sense since: 1) taxane-based chemotherapy induces apoptosis, which is the major source for cfDNA release and; 2) patients with more advanced disease, in later lines of treatment, are more frequently treated with chemotherapy. Therefore, it will be worth to explore and add to the manuscript some figures (as the ones included in the NMed rebuttal letter) reflecting the effect that ARPI or Taxanes have in both cfDNA and ctDNA concentration and ctDNA fraction. How could this affect the prediction model?

While the Reviewer raises a very intriguing and relevant question about the effect of prior treatment on ctDNA%, we feel that the way taxane chemotherapy is applied in clinical populations unfortunately precludes a robust analysis of potential interaction effects. Below are two examples to illustrate these limitations:

1. In our cohort which accrued over a period of time when treatment intensification was not uniformly administered to patients, mCSPC treatment intensification via docetaxel was typically only offered to patients with highly clinically-aggressive disease—as evaluated by their treating physician—whereas patients with more indolent disease were offered ADT monotherapy. This selection bias makes it challenging to determine whether differences in prior treatment exposure affect tumor-intrinsic or -extrinsic determinants of ctDNA% at subsequent timepoints (and consequently the impact of ctDNA% for prognosticating outcomes on future lines of therapy). Overall, the apparent correlation between prior mCSPC taxane intensification and baseline mCRPC ctDNA% is likely entirely explained by clinical circumstance rather than a genuine biological interaction between treatment and ctDNA%-dynamics.
2. Most patients in our metacohort who were treated with first line taxane in the mCRPC setting were enrolled on the OZM-054 trial (NCT02254785), whose enrollment criteria enriched for poor prognosis features (in contrast with the other two cohorts that collectively comprise our study population). Therefore, differences in ctDNA% (measured at baseline or progression) among mCRPC patients exposed to 1L taxane chemotherapy are largely attributable to the confounder of patient/trial selection.

Importantly, these challenges of patient selection bias generalize to all analyses investigating interactions between ctDNA% and treatment exposure, independent of

timepoint and treatment-context. Testing whether specific prior treatment exposure affects subsequent ctDNA% levels ultimately requires analysis of a prospectively enrolled and clinically-standardized patient population that is randomly allocated to different treatments. Our metacohort lacks these necessary conditions. By contrast, treatment exposure within our metacohort is entirely influenced by physician choice, clinical circumstance, and/or trial inclusion testing different SOC (i.e. non-randomized treatment allocation, exemplified in #1 above). In addition, our metacohort includes three distinct patient populations selected using disparate clinical inclusion criteria (i.e. clinically non-standardized, exemplified in #2 above). It is not possible to completely retroactively control for these significant sources of bias.

We have expanded on these limitations in our revised Discussion (new text underlined):

“Fourth, our study contained relatively few patients receiving first- or second-line taxane chemotherapy, with most chemotherapy-treated patients sourced from a single clinical trial enriched for poor prognosis features¹. Small numbers and risk of selection bias precluded examination of potential interactions between treatment class (e.g. ARPI versus taxane) and ctDNA% as a prognostic biomarker. It is plausible that differences in prior treatment exposure may modulate tumor-intrinsic or -extrinsic determinants of ctDNA% at future timepoints, as well as the effect size of ctDNA% for prognosticating subsequent lines of therapy. Furthermore, ctDNA% may have subtly varying prognostic significance for different classes of subsequent treatment (i.e. is a predictive biomarker). Analysis of large clinically-standardized randomized cohorts will be required to uncover potential interactions between drug class (and/or mechanism of action) and ctDNA%-based prognostication. Importantly, the prognostic or predictive implications of ctDNA% remain largely undefined in the context of recent additions to the mCRPC therapeutic armamentarium (e.g. PARP inhibitors and Lutetium-177–PSMA-617 radioligand therapy).”

With respect to our machine-learning tool: for the reasons outlined above, we decided not to include prior treatment exposure as an input feature in our ctDNA% prediction model. Since prior treatment exposure is tied to irrelevant factors related to metacohort composition, it is likely that a gradient-boosting algorithm may memorize this information and its biases, limiting the model’s generalizability to other patient populations. Fortunately, if prior choice of therapy was *influenced* by perceived patient prognosis and disease aggression, this information would already be included as model input features and incorporated into the prediction (since our model leverages direct measurements of prognosis e.g. LDH, ALP, ECOG, PSA). Similarly, if prior treatment exposure improves overall prognosis for *subsequent* lines of treatment, we would also expect this to be captured by the prognostic markers already utilized by our model. The only scenario where it would be necessary to include prior therapy as a model input feature would be if prior treatment *decouples* the correlation between established prognostic markers and ctDNA%, which is not currently known and can only be discovered from a randomized design.

Incidental to the discussion above—for the Reviewer’s interest it is worth noting that treatment induced tumor cell apoptosis likely does not influence ctDNA% at the timepoints measured in our study. While it is plausible that effective anticancer therapy may cause a transient spike in ctDNA% in the minutes to hours after treatment initiation (*N.B.* this has not been demonstrated), ctDNA% fraction rapidly declines within days^{2–5}, but typically resurges surrounding time of clinical progression. Because we are collecting cfDNA *at progression*, the majority of ctDNA likely originates from rapidly

proliferating tumor cells that are no longer responding to therapy. Even if it were true that the majority of progression-timepoint ctDNA arose from treatment induced tumor cell apoptosis, ADT monotherapy also induces strong clinical responses and tumor apoptosis, arguing against the possibility that the quantity of treatment induced apoptotic ctDNA release could be treatment-class specific.

As one would expect, cfDNA concentration and ctDNA fraction tends to be higher in 2L and 3L samples (Figure 1F-H), however, the authors show that the prediction model also works efficiently in this context. How does the model account for ctDNA variance due to different therapies or previous lines of treatment if these variables were not incorporated in the model?

As described in our previous response, we omitted prior treatment as a model input feature to mitigate the possibility of XGBoost incorporating incidental and non-generalizable training cohort characteristics into its predictions. This has the additional important benefit of futureproofing our tool to changes in SOC for metastatic prostate cancer.

In our metacohort, differences in average ctDNA% and cfDNA concentration between mCRPC treatment lines were extremely modest (effect size for ctDNA%: $\eta^2 = 0.006$ [anything <0.01 is considered very small]; $p=0.03$, Kruskal–Wallis one-way analysis of variance). This is compatible with our observation that per-patient ctDNA% remains relatively stable across successive lines of therapy. Nevertheless, we believe that differences in ctDNA% per treatment line are attributable to the fact that later-stage mCRPC tends to be more clinically aggressive and/or have a higher volume of metastatic disease. This can be appreciated in a new Supplementary Figure (shown below) describing the per-line distributions of additional clinical prognostic markers PSA, LDH, ALP, albumin, and hemoglobin, which roughly mirror those of ctDNA% (i.e. higher ctDNA% correlating with an enrichment for poor prognostic factors).

A

- The authors demonstrate in several figures the tight association between cfDNA concentration and ctDNA fraction. In fact, they claim that cfDNA concentration seems to be the highest predictor of ctDNA fraction and cfDNA concentration is inter-correlated with other blood biomarkers such as LDH or ALP. However, they do not explore the association between cfDNA concentration and other clinical parameters (metastases, time to progression, etc.) or outcome. What would be the prognostic role of cfDNA concentration? How would it compare to ctDNA?

Our original submission included most of the analyses the Reviewer is requesting, although we appreciate there is opportunity to better signpost these in the manuscript text, as well as more extensively investigate associations between cfDNA concentration and baseline clinical characteristics. We had initially tested whether cfDNA concentration more strongly predicts overall survival than ctDNA fraction in both the first- and second-line setting—relevant Results paragraph (key parts underlined) and original Supplementary Figure pasted below:

Results: “We additionally tested whether ctDNA concentration (i.e. nanograms of ctDNA per mL plasma, the product of total cfDNA concentration and ctDNA%) enabled more precise prognostication than ctDNA% or cfDNA concentration alone. When dichotomized

by median, ctDNA% and ctDNA concentration were associated with comparable univariable hazard ratios for OS (HR=3.18 [95% CI: 2.53-3.99], p<0.001; HR=3.28 [95% CI: 2.61-4.12], p<0.001) and both enabled superior patient stratification relative to cfDNA concentration (HR=2.05 [95% CI: 1.64-2.56], p<0.01) (Fig 4a; Supplementary Fig 6)”

In addition, our original Supplementary Table 4 contained univariate hazard ratios for ctDNA% (dichotomized by median) and cfDNA concentration (dichotomized by median) for PSA-PFS and OS in both the first- and second-line settings (i.e. 8 survival analyses total) – key statistics summarized below:

Clinical context and endpoint	cfDNA concentration HR	ctDNA% HR
------------------------	-----------

First-line PSA-PFS	1.65	2.80
First-line OS	2.05	3.18
Second-line PSA-PFS	1.25 [not significant]	1.65
Second-line OS	1.79	2.51

Collectively these analyses indicate that cfDNA concentration dichotomized at median consistently results in weaker prognostic stratification compared to ctDNA fraction, regardless of endpoint (PFS or OS) or line of treatment (first- or second-line). We have better clarified this in a new Results line:

“ctDNA% was more strongly prognostic than cfDNA concentration (both variables dichotomized at median) independent of treatment line and endpoint (Fig 4a,b; Supplementary Fig 6; Supplementary Table 4)”

To provide deeper biological granularity into the link between cfDNA concentration and prognostic clinical variables, we have also generated a new Supplementary Figure showing the association between cfDNA concentration and seven prognostic clinical indices (analogous to Figure 2).

New Results text linked to the Figure above: “cfDNA concentration was similarly correlated with most aforementioned clinical factors, although the effect size was weaker relative to ctDNA% (Supplementary Fig 4; Fig 2c).”

- Previous studies showed that the amount of input blood (ml of plasma) used for cfDNA isolation could affect cfDNA concentration yield (Alborelli et al., 2019, Cell Death Dis.). Could this also affect ctDNA fraction? If so, how could this affect the predictions of the model?

The study the Reviewer is referring to (Alborelli et al., 2019, Cell Death Dis.; Figure 1b-c) shows that total cfDNA yield (in nanograms) is expectedly proportional to plasma volume used for extraction. cfDNA concentration (i.e. nanograms per mL of plasma) is not affected by plasma input volume.

However, sufficient input DNA quantity is important for generating successful libraries. Libraries made from small amounts of DNA (roughly <10ng) typically contain more PCR duplicate fragments and have lower molecular diversity, resulting in increased sample noise that impedes variant-detection sensitivity and accuracy⁶. ctDNA fraction estimates in samples using low cfDNA input will be less reliable.

Fortunately, most commercial and clinical ctDNA genotyping targeted assays (e.g., FoundationOne Liquid CDx) conservatively require 20-30ng of cfDNA for library construction (our study used 25ng libraries). The minority of samples (approximately <5%) with insufficient cfDNA yield are considered QC-fail and are not subjected to sequencing. In practice, strict quality control during sample processing and library preparation means that low cfDNA yield does not affect assay performance or ctDNA fraction estimation accuracy.

Reassuringly, validation of our parsimonious 8-variable model using the ProBio dataset—which utilized different preanalytical and bioinformatic methodology to our study—achieved a nearly identical AUC to our metacohort (0.76 vs. 0.78, respectively). This largely implies that our model is generalizable to any sufficiently optimized ctDNA clinical genotyping protocol.

Minor comments:

- In Figure 1B, the number of patients with bone mets does not match the percentages shown in Figure 1D.

Thank you for pointing this out. Most metacohort patients were evaluated for presence/absence of bone lesions, including patients providing cfDNA samples in the second- and third-line context (Figure 1D). However, we only reviewed imaging data to *enumerate* bone lesions for patients at first-line (shown in Figure 1B), explaining the apparent discrepancy between these two Figures.

We have added the following line to the Figure 1b legend to clarify:

“Note that bone metastases were only enumerated in the first-line context, although all patients (independent of treatment line) were evaluated for bone lesion presence/absence....”

If helpful to the Reviewer, in our original submission we included a Supplementary Table detailing the extent of missing clinical annotation for all clinical fields per line of treatment, including variables of ‘bone metastases (presence/absence)’ and ‘Number of bone metastases’.

- In Figure 4G the y-axis legend is a bit confusing. Less ctDNA fraction is associated with a better PSA response (more patients with more than 50% PSA reduction), however, this is not completely clear with the representation shown in here.

We have adjusted our Methods to clarify how best PSA response was calculated (new text underlined): “PSA response was defined as $\geq 50\%$ PSA decline from the baseline pretreatment measurement, calculated using the on-treatment PSA nadir (standard PCWG2 criteria)”.

The Figure 4g caption also may help clarify the y-axis legend to readers: “(g) Waterfall plot showing best PSA response (relative to baseline PSA) on first-line mCRPC therapy stratified by baseline ctDNA% (ctDNA>30%, ctDNA 2-30%, and ctDNA<2%)”. This waterfall plot was modeled after a relatively common visualization utilized in other prostate cancer clinical and translational studies (e.g. Annala et al., 2021, Annals of Oncology [PMID: 33836265] Figure 2b; Buteau et al., 2023, Lancet Oncology [PMID: 36261050], Figure 1; Azad et al., 2015, European Urology [PMID: 25018038], Figure 1).

Reviewer #5 (Remarks to the Author):

In this well-written manuscript, the authors provide a detailed analysis of the clinical determinants of circulating tumor DNA fraction (ctDNA%) and its utility for prognostication using a large metacohort of 491 mCRPC patients. Using this data, they then develop a machine learning-based tool that predicts the likelihood that a patient will have a sufficiently high ctDNA% for clinically informative ctDNA genotyping. The study highlights ctDNA% as a validated tool for patient risk stratification, and provides a potentially clinically useful web-based tool for estimating ctDNA% using clinical parameters, prior to actual ctDNA testing.

The authors have done an excellent job responding to the Reviewer comments. In particular, they are to be commended for performing validation studies of their ctDNA% prediction model with two additional external cohorts. The revised text also nicely clarifies the intended clinical application of their ctDNA% prediction machine learning-based tool.

Thank you for the supportive comments and the helpful feedback.

I have the following additional minor comments:

Line 209; Fig. 3F – Can the authors speculate on the potential nature of the additional patient- or tumor-specific determinants of ctDNA% that are not included in their models?

There are numerous additional variables that could hypothetically affect patient ctDNA% that are not accounted for by our XGBoost prediction model. ctDNA% is thought to mostly reflect total tumor burden and innate tumor-cell properties (e.g. proliferative capacity and therefore clinical aggression), but can also be modulated via a variety of tumor-extrinsic physiological factors (N.B. many factors are probably impossible to assess *a priori*). The features incorporated into our 18-variable model were mainly metrics of tumor burden (e.g. number of bone lesions, LDH, ALP) and prognosis, since these are relatively easily measured. It is possible that inclusion of features that more directly measure tumor cell *proliferation* may improve model ctDNA%-prediction

accuracy. For example, leveraging variables such as tumor metabolic activity (e.g. total lesion glycolysis via [¹⁸F]FDG-PET/CT) or percent of tumor nuclei positive for Ki-67 as surrogates for tumor cell proliferation. Additionally, it may be relevant to explore whether the local tumor microenvironment constrains ctDNA release. For example, ctDNA release from metastases with a high degree of macrophage infiltration may be comparatively limited (i.e. due to immune cell-mediated phagocytosis preventing release of post-apoptotic tumor cell detritus into circulation). Finally, it is likely that certain genomic alterations (as indicators of tumor aggression) may also impact ctDNA%.

Although our models already incorporated metrics of total tumor burden and anatomic involvement, we would also posit that more granular evaluation of these variables would also improve ctDNA% model prediction accuracy. New next-generation imaging tools (e.g. [⁶⁸Ga]PSMA-PET/CT and [¹⁸F]FDG-PET/CT in prostate cancer) can provide highly quantitative estimates of total tumor volume, outperforming the conventional imaging analysis utilized in this study.

We have added a new line to our Discussion to comment on other possible modulators of ctDNA%:

“New studies investigating additional determinants of ctDNA% should utilize next-generation targeted imaging (e.g. [⁶⁸Ga]PSMA-PET/CT in prostate cancer) for more precise quantification of disease burden and location—as well as investigate the potential relevance of tumor cell proliferation indicators (e.g., Ki-67-positive tumor nuclei or total lesion glycolysis) and microenvironmental factors (e.g., tumor vascularization, macrophage infiltration) on ctDNA%”

Lines 239-241 – Please provide Tables showing the summary clinical characteristics of the two external prospective mCRPC datasets used for validation.

Clinical characteristics for the OPT/ILU validation cohort (representing a pooled analysis of patients from the OPTIMUM (NCT02426333) and ILUMINATE (NCT02471469) prospective trials) has been published previously—see Table 1 from Tolmeijer et al., *Clinical Cancer Research* 2023; PMID: 36996325². To avoid duplication of previously published data, we have amended our Results text to explicitly refer readers to this publication:

- **Results (new text underlined):** *“We validated the performance of our parsimonious 8 feature model in two external prospective mCRPC datasets collectively including 391 patients with first-line mCRPC, achieving similar AUCs for predicting ctDNA \geq 2% of 0.76-0.78 (Methods; Fig 3g-h, Supplementary Fig 4, Supplementary Table 6). Patient clinical characteristics for one of the two validation cohorts (n=81 patients) has been published previously (Tolmeijer et al., *Clinical Cancer Research* 2023).”*

For the second validation cohort (ProBio trial: NCT03903835), select patient demographic details can be gleaned from the trial clinical inclusion/exclusion criteria: available at <https://clinicaltrials.gov/study/NCT03903835> and two recent publications dissecting the trial design (Crippa et al., 2020, PMID: 32586393; De Laere et al., 2022, PMID: 35317973).

ProBio largely focuses on the first- and second-line mCRPC setting, and cohort clinical characteristics will be described in further detail in an upcoming publication.

Line 266 – Please delete the extra period.

We have corrected this typo.

Line 279; Fig. 4A and 4D – The labeling of the X-axis is not clear and is not consistent with text in the Figure Legend. Should this be “Time from 1L therapy initiation to death” rather than “mCRPC diagnosis to death”?

We have now corrected the Figure 4A and 4D x-axis label to read “1L treatment initiation to death (months)”, consistent with the original figure caption and clinical endpoints description in the Methods.

Lines 426-429; Table 1 – Please add race as a clinical characteristic and provide a % breakdown of the racial distribution of patients within the metacohort and validation cohorts. The authors have already described the predominance of European ancestry patients as a limitation of their study, but it would still be useful to have the actual percentages presented in the Tables.

We do not routinely collect self-reported race or other measures of ancestral genetic background. Our statement about the predominance of European ancestry is an assumption based on our experience of the typical metastatic prostate cancer patient treated in Canada (main metacohort) and Northern Europe (ProBio validation cohort). Ethnicity data for the pooled OPTIMUM (NCT02426333) and ILLUMINATE (NCT02471469) validation cohort has been published previously (see Table 1 from Tolmeijer et al., Clinical Cancer Research 2023; PMID: 36996325)².

We have now modified our limitation sentence to point out that we did not collect race as a characteristic: “Finally, although we did not collect self-reported race or other measures of patient genetic background, based on the demographics of the jurisdictions contributing to our metacohort and validation cohorts we can assume that patients were primarily of European ancestry.”

Reviewer #6 (Remarks to the Author):

The manuscript by Fonseca et al. describes the development of a novel tool for decision-making on whether cfDNA-based genotyping is viable based on the probability of ctDNA detection. In addition, they demonstrate the prognostic value of ctDNA levels in patients with advanced prostate cancer. The latter, while not entirely novel, it underscores previous work using a large meta-cohort, in which multiple confounders could be evaluated to demonstrate ctDNA% as an independent predictive biomarker.

Overall, the manuscript is very clear, and the study and analyses are well described. A major comment relates to Figure 6, which summarises the potential clinical pipeline using the developed tool. However, from the figure, it is not apparent that the ctDNA prediction tool is the 2nd in the decision-making process. The fact that the arrow goes from patients to the tool, implies that it is the 1st step.

Thank you for the time you have taken to perform this peer-review task and for the constructive feedback.

Our rationale for placing the ctDNA%-prediction tool *first* in the workflow was that in the event that routine ctDNA-testing is available, ctDNA.org can help users determine whether to pursue simultaneous tissue genotyping in case predicted ctDNA% is low. ctDNA-testing provides important prognostic information (in the form of ctDNA%) regardless of adequacy for genotyping, and therefore should ideally be performed for all patients if feasible.

However we recognize that this intended decision tree was not sufficiently clear in the original Figure 5. We have now adjusted Figure 5:

I just have some minor suggestions and queries:

1. While the changes to 'ctDNA biomarker genotyping' provide some clarity, it is not always appropriate. Such as, in this sentence in page 3: "Excitingly, ctDNA% is increasingly reported on commercially-available ctDNA biomarker genotyping tests 23, meaning that ctDNA%-prognostication is poised to rapidly influence patient management pending its clinical validation."

We've updated the above sentence to "*Excitingly, ctDNA% is increasingly reported on commercially-available tests that genotype ctDNA to determine treatment-predictive biomarker status, meaning that ctDNA%-prognostication is poised to rapidly influence patient management pending its clinical validation*". We also checked the manuscript and made one further text clarification related to ctDNA genotyping.

2. Page 7: 'credential' is not a verb

We have modified this sentence to the following:

"Our data, together with prior smaller studies, authenticate ctDNA%..."

3. Methods: There are no details on the supplier companies, country, etc, which is expected as per most journal guidelines.

We have added these details to the methods.

4. Methods, page 17: The way blood processing is described, sounds like only Streck derived plasma samples were re-spun and used for buffy coat collection.

We have updated these sentences, thank you.

5. Methods, page 18, when referring to “ctDNA% = $2/(1/VAF + 1)$ ” and also “ctDNA% = $2 - VAF - 1$ ”, I believe you are implying an adjustment factor rather than the %ctDNA itself. Please describe clearly. Otherwise, irrespective of VAF, all ctDNA values will fall below 2.

In general, ctDNA fraction—i.e. the proportion of total cfDNA that is tumor-derived—is calculated from the population prevalence of one or more somatic features detected in cfDNA. The formulae on page 18 are not adjustment factors, but are rather the mathematical relationships between ctDNA fraction and the VAF of alterations directly measured in cfDNA (that are exploited to *infer* ctDNA fraction). These formulae represent standard approaches for calculating ctDNA fraction from targeted sequencing data and have been utilized in established bioinformatic software ^{7,8} and prior papers ^{9–13}.

We recognize that one possible source of confusion in these formulae is our use of the abbreviation ‘ctDNA%’ to refer to ctDNA fraction rather than ctDNA percentage (as the abbreviation erroneously suggests). In other words, ctDNA% in these formulae refer to a quantity between 0 and 1 (rather than 0% and 100%). We adopted this nomenclature for brevity in describing ctDNA fraction throughout the manuscript, and have modified our Methods to clarify (new text underlined):

“In regions of LOH, mutation VAF and ctDNA% (both as variables with lower and upper bounds of 0 and 1, respectively) are mathematically related as ctDNA% = $2/(1/VAF + 1)$ ”

References

1. Annala, M. *et al.* Cabazitaxel versus abiraterone or enzalutamide in poor prognosis metastatic castration-resistant prostate cancer: a multicentre, randomised, open-label, phase II trial. *Ann. Oncol.* 32, 896–905 (2021).
2. Tolmeijer, S. H. *et al.* Early on-treatment changes in circulating tumor DNA fraction and response to enzalutamide or abiraterone in metastatic castration-resistant prostate cancer. *Clin. Cancer Res.* (2023) doi:10.1158/1078-0432.CCR-22-2998.
3. Vandekerkhove, G. *et al.* Circulating Tumor DNA Abundance and Potential Utility in De Novo Metastatic Prostate Cancer. *Eur. Urol.* 75, 667–675 (2019).
4. Jayaram, A. *et al.* Plasma tumor gene conversions after one cycle abiraterone acetate for metastatic castration-resistant prostate cancer: a biomarker analysis of a multicenter international trial. *Ann. Oncol.* 32, 726–735 (2021).
5. Tan, W. *et al.* Dynamic changes in gene alterations during chemotherapy in metastatic castrate resistant prostate cancer. *Sci. Rep.* 12, 4672 (2022).
6. McNulty, S. N., Mann, P. R., Robinson, J. A., Duncavage, E. J. & Pfeifer, J. D. Impact of Reducing DNA Input on Next-Generation Sequencing Library Complexity and Variant Detection. *J. Mol. Diagn.* 22, 720–727 (2020).
7. Loh, J. W. *et al.* All-FIT: allele-frequency-based imputation of tumor purity from high-depth

- sequencing data. *Bioinformatics* 36, 2173–2180 (2020).
8. Prandi, D. & Demichelis, F. Ploidy- and Purity-Adjusted Allele-Specific DNA Analysis Using CLONETv2. *Curr. Protoc. Bioinformatics* 67, e81 (2019).
 9. Renaud, G. *et al.* Unsupervised detection of fragment length signatures of circulating tumor DNA using non-negative matrix factorization. *Elife* 11, (2022).
 10. Orlando, F. *et al.* Allele-informed copy number evaluation of plasma DNA samples from metastatic prostate cancer patients: the PCF_SELECT consortium assay. *NAR Cancer* 4, zcac016 (2022).
 11. Annala, M. *et al.* Circulating Tumor DNA Genomics Correlate with Resistance to Abiraterone and Enzalutamide in Prostate Cancer. *Cancer Discov.* 8, 444–457 (2018).
 12. Mizuno, K. *et al.* Clinical Impact of Detecting Low-Frequency Variants in Cell-Free DNA on Treatment of Castration-Resistant Prostate Cancer. *Clin. Cancer Res.* 27, 6164–6173 (2021).
 13. Kohli, M. *et al.* Clinical and genomic insights into circulating tumor DNA-based alterations across the spectrum of metastatic hormone-sensitive and castrate-resistant prostate cancer. *EBioMedicine* 54, 102728 (2020).

REVIEWERS' COMMENTS

Reviewer #4 (Remarks to the Author):

The authors have thoroughly responded and acknowledged most of the Reviewer comments. My main concern is still the lack of novelty from the prognostication value of the previously published patient cohorts (representing 59% of the presented cohorts). However, I acknowledge that the authors have recognized this crucial aspect and mitigated the emphasis on this particular facet of the study, even going so far as to modify the original title of the manuscript.

The question regarding prior exposure to other treatment lines and is well explained and nicely justified in the updated version of the manuscript (new discussion section + new supplementary figure for line of treatment vs clinical parameters).

In addition, I believe the addition of the comparison of ctDNA% vs cfDNA concentration is an important result that should be also included in this version of the manuscript. Despite ctDNA% moderately outperforms cfDNA concentration for clinical prognostication, the potential value of the latest is worth to acknowledge. The use of cfDNA concentration as a biomarker is independent of sequencing costs and platform access which might simplify the application of this liquid biopsy tool in the clinical practice.

Overall, the updated version of the manuscript provides a clear explanation of the potential of their machine learning-based tool for predicting ctDNA% for clinical application. Additionally, it acknowledges potential limitations and caveats associated with their study.

Reviewer #5 (Remarks to the Author):

The authors have adequately addressed my comments, as well as the comments of the other reviewers in my opinion.

Reviewer #6 (Remarks to the Author):

I am satisfied with the responses provided by the authors and the amended manuscript.

Response to Reviewers

Reviewer #4 (Remarks to the Author):

The authors have thoroughly responded and acknowledged most of the Reviewer comments. My main concern is still the lack of novelty from the prognostication value of the previously published patient cohorts (representing 59% of the presented cohorts). However, I acknowledge that the authors have recognized this crucial aspect and mitigated the emphasis on this particular facet of the study, even going so far as to modify the original title of the manuscript.

The question regarding prior exposure to other treatment lines and is well explained and nicely justified in the updated version of the manuscript (new discussion section + new supplementary figure for line of treatment vs clinical parameters).

In addition, I believe the addition of the comparison of ctDNA% vs cfDNA concentration is an important result that should be also included in this version of the manuscript. Despite ctDNA% moderately outperforms cfDNA concentration for clinical prognostication, the potential value of the latest is worth to acknowledge. The use of cfDNA concentration as a biomarker is independent of sequencing costs and platform access which might simplify the application of this liquid biopsy tool in the clinical practice.

Overall, the updated version of the manuscript provides a clear explanation of the potential of their machine learning-based tool for predicting ctDNA% for clinical application. Additionally, it acknowledges potential limitations and caveats associated with their study.

Thank you for your constructive feedback.

The prognostic relevance of cfDNA concentration is addressed in several places within the Results (specific analyses/figures underlined):

- ***“cfDNA concentration was similarly correlated with most aforementioned clinical factors, although the effect size was weaker relative to ctDNA% (Supplementary Figure 4; Fig 2c).”***
- ***“ctDNA% was more strongly prognostic than cfDNA concentration (both variables dichotomized at median) independent of treatment line and endpoint (Fig 4a,b; Supplementary Figure 8; Supplementary Table 4).”***

We have now amended our Discussion to explicitly acknowledge the prognostication implications of cfDNA concentration (new text underlined):

“... We believe that the ctDNA% risk categories validated herein (low, medium, and high) provide a useful working model for the immediate clinical implementation of ctDNA% as a prognostic aide in mCRPC. cfDNA concentration was also prognostic for outcomes, although the effect size (i.e. hazard ratio) was consistently smaller than ctDNA% (Supplementary Figure 8; Supplementary Table 4). In theory, the weaker prognostic stratification via cfDNA concentration may be pragmatically offset by the

relative expediency and lower cost of quantification (compared to ctDNA%); however, ctDNA sequencing has the notable advantage of enabling simultaneous prognostication (via ctDNA%) and analysis of prognostic and treatment-predictive genomic alterations. Integrating ctDNA% with somatic information (e.g. TP53 defects) may provide additional prognostic granularity albeit potentially at the cost of added implementational complexity.

Reviewer #5 (Remarks to the Author):

The authors have adequately addressed my comments, as well as the comments of the other reviewers in my opinion.

Thank you.

Reviewer #6 (Remarks to the Author):

I am satisfied with the responses provided by the authors and the amended manuscript.

Thank you.